# Edit Flows: Variable Length Discrete Flow Matching with Sequence-Level Edit Operations

**Marton Havasi**
FAIR at Meta

**Brian Karrer**
FAIR at Meta

**Itai Gat**
FAIR at Meta

**Ricky T. Q. Chen**
FAIR at Meta

## Abstract

Autoregressive generative models naturally generate variable-length sequences, while non-autoregressive models struggle, often imposing rigid, token-wise structures. We propose Edit Flows, a non-autoregressive model that overcomes these limitations by defining a discrete flow over sequences through edit operations—insertions, deletions, and substitutions. By modeling these operations within a Continuous-time Markov Chain over the sequence space, Edit Flows enable flexible, position-relative generation that aligns more closely with the structure of sequence data. Our training method leverages an expanded state space with auxiliary variables, making the learning process efficient and tractable. Empirical results show that Edit Flows outperforms both autoregressive and mask models on image captioning and significantly outperforms the mask construction in text and code generation.

## 1 Introduction

Non-autoregressive models have become the standard across high-dimensional modalities, thanks to their ability to produce coherent and globally consistent outputs. Recent advances include MovieGen (Polyak et al., 2025) for video, Audiobox (Vyas et al., 2023) for audio, and Stable Diffusion 3 (Esser et al., 2024) for images. This trend extends to discrete code and text generation as well: recent diffusion-based models such as LLaDa (Nie et al., 2025), DREAM (Ye et al., 2025), and Mercury (Ermon et al., 2025) show that fully parallel generation can match or even surpass strong autoregressive baselines on certain open-ended language tasks. Despite these advances, current non-autoregressive models rely on rigid, factorized representations with fixed token positions. They work by iteratively unmasking or replacing tokens in the target sequence. Critically, they cannot add or remove tokens: two fundamental operations for modeling sequential data.

In this paper, we propose *Edit Flows*, a novel non-autoregressive framework that models generation as a discrete flow over the space of sequences via *edit operations*—insertions, deletions, and substitutions. We frame sequence generation as a stochastic process governed by a Continuous-time Markov Chain (CTMC) over full sequences, in contrast to the usual factorized representation with absolute token positions (Figure 1). The model learns to estimate the rate of each possible edit operation conditioned on the current sequence (Figure 2). This enables modeling based on relative token positions and eliminates the need for masking or padding tokens during training or inference. Moreover, Edit Flows naturally accommodate variable-length sequences. In contrast to existing non-autoregressive models that generate tokens in fixed lengths or rely on heuristic semi-autoregressive sampling (Nie et al., 2025), Edit Flows can produce longer or shorter outputs adaptively, depending on the context.

Despite the conceptual simplicity of modeling sequence transitions through edits, training such models is non-trivial. A direct optimization of full sequence-level stochastic processes typically demands costly computations. To address this, we introduce a Flow Matching-based (Lipman et al., 2024) training procedure that augments the state space with auxiliary variables that determine one possible chain of edits that leads to the target sequence. By sampling these auxiliary variables in

39th Conference on Neural Information Processing Systems (NeurIPS 2025).

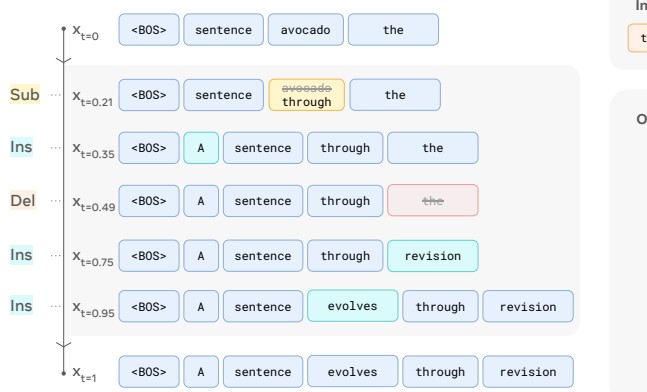
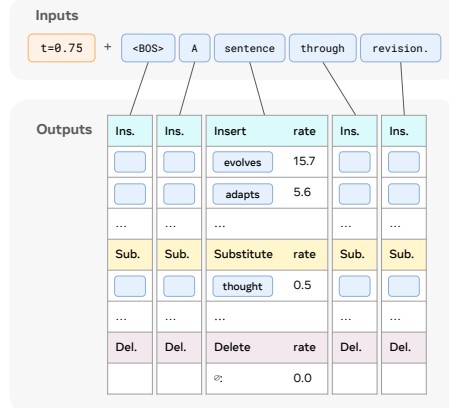

Figure 1: **Edit Flow sampling process.** Starting with $x_0$ containing random tokens or an empty sequence, the model applies edits to $x_t$ and reaches a cohesive sentence at time $t = 1$.

Figure 2: **Edit Flow model inputs and outputs.** Given $x_t$, the model predicts the rate of each possible edit.

each training iteration (without exposing them to the model), we obtain a tractable training objective and the model automatically learns to infer these auxiliary variables.

Empirically, Edit Flows show a strong and consistent improvement over fixed-length discrete flow and diffusion models (Campbell et al., 2024b; Gat et al., 2024; Shi et al., 2024) across several benchmarks, including image-to-text generation at 280M parameter scale (MS-COCO, Image Captioning 3M), code generation at 1.3B parameter scale (HumanEval, MBPP), and open-ended text benchmarks at 1.3B parameter scale (HellaSwag, ARC, PIQA, OBQA, WinoGrande). On image-to-text generation, Edit Flows outperformed all baselines, including the autoregressive model, and on code generation, it has a relative improvement of 138% over the mask model. We summarize our contributions:

▷ We introduce Edit Flows, a non-autoregressive generation framework expanding upon the Discrete Flow Matching recipe, with native support for variable-length sequences via edit operations—insertions, substitutions, and deletions.

▷ We construct a sequence-level probability path, enabling CTMC-based modeling directly over sequences of varying lengths, unlike prior work focused on token-level transitions.

▷ We demonstrate the effectiveness of Edit Flows on large-scale benchmarks in image captioning, open-ended text benchmarks, and code generation.

## 2 Preliminaries

### 2.1 Continuous-time Markov Chains

To form the basis of our discrete generative model (Campbell et al., 2024b; Gat et al., 2024; Holderrieth et al., 2024; Shaul et al., 2024), we make use of Continuous-time Markov Chains (CTMC) over a discrete space $\mathcal{X}$. These are Markov processes that generate trajectories $(X_t)_{t \in [0,1]}$ and is characterized by a *rate* $u_t$ denoting the infinitesimal transition probabilities between states

$$\mathbb{P}(X_{t+h} = x | X_t = x_t) = \delta_{x_t}(x) + h u_t(x|x_t) + o(h) \tag{1}$$

where $o(h)$ satisfies $\lim_{h \to 0} \frac{o(h)}{h} = 0$n. Sampling from a CTMC can be done by iteratively applying the update formula (1). The rate $u_t(x|x_t)$ denotes the infinitesimal probabilities of transitioning from a state $x_t$ to any other state $x$ at time $t$, and for (1) to be a proper probability mass function, we need both sides to sum to one. Hence, $u_t$ needs to satisfy

$$u_t(x|x_t) \geq 0 \text{ for all } x \neq x_t, \quad \sum_x u_t(x|x_t) = 0, \tag{2}$$

typically referred to as the *rate conditions*. Note this enforces $u_t(x_t|x_t) = -\sum_{x \neq x_t} u_t(x|x_t)$.

We say a rate $u_t$ "generates" a probability path $p_t$ if the time marginals of the associated CTMC are samples from $p_t$, *i.e.*, $X_t \sim p_t$. Concretely, they should satisfy the Kolmogorov forward equation,

$$\frac{\partial}{\partial t} p_t(x) = \sum_y u_t(x|y) p_t(y) = \underbrace{\sum_{y \neq x} u_t(x|y) p_t(y)}_{\text{flow into } x} - \underbrace{\sum_{y \neq x} u_t(y|x) p_t(x)}_{\text{flow out of } x}. \quad (3)$$

That is, the change in probability of being in state $x$ is the total infinitesimal probability flowing into $x$ from other states minus the total infinitesimal probability flowing out of $x$, determined by the rate.

## 2.2 Discrete Flow Matching

Discrete Flow Matching (DFM; Campbell et al. 2024b; Gat et al. 2024) is a conceptually simple framework for learning a CTMC-based generative model to transport from a source (*e.g.* noise) distribution $p(x)$ to a target (*e.g.* data) distribution $q(x)$ over a discrete space $x \in \mathcal{X}$. For now, consider a discrete space over sequences of fixed length $N$, so $\mathcal{X} = \mathcal{T}^N$ where $\mathcal{T} = \{1, \dots, M\}$ denotes a vocabulary of size $M$ containing a discrete set of token values.

Discrete FM training relies on prescribing a *coupling* distribution $\pi(x_0, x_1)$ that samples pairs $(x_0, x_1)$ where the marginals are $p$ and $q$, *i.e.*,

$$\sum_{x_0} \pi(x_0, x_1) = q(x_1), \qquad \sum_{x_1} \pi(x_0, x_1) = p(x_0). \quad (4)$$

The simplest case is of course the independent coupling $\pi(x_0, x_1) = p(x_0) q(x_1)$. Further, we would also prescribe a *conditional* CTMC characterized by a conditional rate

$$u_t(x|x_t, x_0, x_1) \text{ generating } p_t(x|x_0, x_1), \text{ s.t. } p_0(x|x_0, x_1) = \delta_{x_0}(x), \ p_1(x|x_0, x_1) = \delta_{x_1}(x) \quad (5)$$

where $\delta$ denotes Kronecker's delta function. That is, the conditional probability path $p_t(x|x_0, x_1)$ interpolates between two *points* from the source and target. DFM then trains a generative model that transports according to the marginal probability path $p_t(x)$, which interpolates between the source and target *distributions*.

$$p_t(x) = \sum_{x_0, x_1} p_t(x|x_0, x_1) \pi(x_0, x_1) \quad \text{implying} \quad p_0(x) = p(x), \ p_1(x) = q(x). \quad (6)$$

It can be shown that the *marginal* rate

$$u_t(x|x_t) = \mathbb{E}_{p_t(x_0, x_1|x_t)} u_t(x|x_t, x_0, x_1) \quad (7)$$

generates the marginal probability path $p_t(x)$, *i.e.* $u_t(x|x_t)$ characterizes a CTMC that transports from the source $p$ to the target data distribution $q$. In order to train a model to approximate (7), prior works have used cross-entropy (Gat et al., 2024; Campbell et al., 2024b) and evidence lower bounds (Lou et al., 2024; Sahoo et al., 2024; Shi et al., 2024; Shaul et al., 2024) as training objectives, all of which are captured by the family of Bregman divergences (Holderrieth et al., 2024).

**Token-wise mixture paths.** The prescription of (4) and (5) is then left as a design choice. Most existing works have focused on the factorized *token-wise* conditional path (Gat et al., 2024)

$$p_t(x^i|x_0^i, x_1^i) = (1 - \kappa_t) \delta_{x_0^i}(x^i) + \kappa_t \delta_{x_1^i}(x^i), \quad u_t(x^i|x_t^i, x_0^i, x_1^i) = \frac{\dot{\kappa}_t}{1 - \kappa_t} \left( \delta_{x_1^i}(x^i) - \delta_{x_t^i}(x^i) \right), \quad (8)$$

where $\kappa_t$ is a scheduler that satisfies $\kappa_0 = 0$, $\kappa_1 = 1$. The multi-dimensional case is to consider only states that differ by one token, expressed concisely as

$$p_t(x|x_0, x_1) = \prod_{i=1}^N p_t(x^i|x_0^i, x_1^i), \quad u_t(x|x_t, x_0, x_1) = \sum_i \delta_{x_t}(x^{\neg i}) u_t(x^i|x_t^i, x_0^i, x_1^i), \quad (9)$$

where $\delta_{x_t}(x^{\neg i}) = \prod_{j \neq i} \delta_{x_t^j}(x^j)$ is a shorthand for denoting that all dimensions except $i$ are the same. That is, this rate is *factorized* in that it only describes token-wise changes, though sampling can be done in parallel (1). This is a particular advantage of using a continuous-time framework, requiring only a per-dimension parameterization of the model, at the cost of using an iterative procedure for sampling. It has been difficult to generalize beyond the token-wise paths as it can quickly become intractable to prescribe a conditional CTMC (5) for training that has more general transitions over sequence space (Shaul et al., 2024).

**Mask construction.** As noted by many existing works (Austin et al., 2021; Lou et al., 2024; Campbell et al., 2024b), the simplifying case of considering the source distribution to be a mask

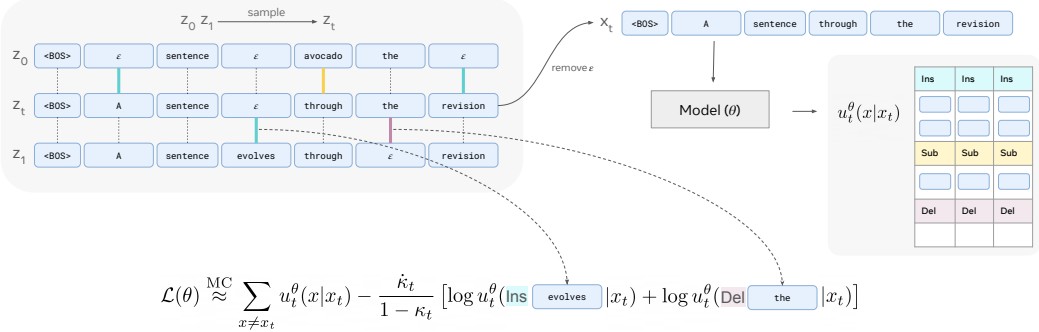

$$\mathcal{L}(\theta) \stackrel{\text{MC}}{\approx} \sum_{x \neq x_t} u_t^\theta(x|x_t) - \frac{\dot{\kappa}_t}{1 - \kappa_t} \left[ \log u_t^\theta(\text{Ins } \boxed{\text{evolves}} | x_t) + \log u_t^\theta(\text{Del } \boxed{\text{the}} | x_t) \right]$$

Figure 3: Computing the loss starts with the two aligned sequences $z_0$ and $z_1$. Locations where $z_0^i = \varepsilon$ require an insertion operation, locations where $z_1^i = \varepsilon$ require a deletion and locations where $z_0^i \neq z_1^i$ require a substitution. $z_t$ is sampled by applying a subset of the operations to $z_0$ depending on the scheduler. Then, $x_t$ is obtained by removing all $\varepsilon$ tokens from $z_t$. The Monte-Carlo estimate of the loss contains the model output $u_t^\theta(x|x_t)$ in two terms: the negated sum of all the edit rates and the logarithms of the remaining edits between $z_t$ and $z_1$.

distribution has significant theoretical and practical benefits. That is, setting $p_0(x) = \delta_{\mathbb{m}}(x)$, where $\mathbb{m}$ is a special *mask* token not found in the original vocabulary. Theoretically, this drastically simplifies the construction (Sahoo et al., 2024; Shi et al., 2024) and practically has been shown to scale (Nie et al., 2025; Ye et al., 2025; Ermon et al., 2025). The main benefits come from requiring only learning transitions between the mask token and the other tokens, with no transitions between tokens from the original vocabulary. However, this construction still has multiple downsides, as it does not make full use of the CTMC framework and is equivalent to an any-order autoregressive model (Hoogeboom et al., 2022; Pannatier et al., 2024) though usually implemented with non-causal attention. As with all token-wise path constructions, the most glaring downside is the lack of inherent support for variable-length generation. To handle variable length outside of the modeling framework, padding can be done during training but the excessive padding makes the model over-confident in predicting padding tokens, an issue that currently relies on semi-autoregressive sampling to get around (Nie et al., 2025).

## 3 Edit Flows

### 3.1 Edit Flows: a continuous-time Markov chain using edit operations

We design a new CTMC-based generative model through the Discrete Flow Matching framework using edit operations to enable variable length generation, while encompassing existing constructions as special cases. Let $\mathcal{T}$ be as defined previously to be a vocabulary of size $M$. Then our state space is defined as the set of all possible sequences up to some maximum length $N$, *i.e.*, $\mathcal{X} = \bigcup_{n=0}^{N} \mathcal{T}^n$.

We will now describe the Edit Flow model which is a CTMC that operates directly on the space of sequences, and discuss tractable training using a generalization of the DFM recipe later in Section 3.2. Specifically, we parameterize the rate of a CTMC $u_t^\theta$. For two sequences $x, x_t \in \mathcal{X}$, $u_t^\theta(x|x_t)$ is allowed non-zero only if $x$ and $x_t$ differ by one *edit operation*. An edit operation is one of either *insertion*, *deletion*, or *substitution*, which we use to transition between sequences in our generative model. Specifically, given a sequence $x$ with variable length $n(x)$, we define the edit operations that can be performed on $x$ concretely as follows.

▶ Let $\text{ins}(x, i, a)$, $x \in \mathcal{X}, i \in \{1, \ldots, n(x)\}, a \in \mathcal{T}$, be the sequence resulting from inserting the token value $a$ to the right side of position $i$ of the sequence $x$, resulting in
$$\text{ins}(x, i, a) = (x^1, \ldots, x^i, a, x^{i+1}, \ldots, x^{n(x)}). \tag{10}$$

▶ Let $\text{del}(x, i)$, $x \in \mathcal{X}, i \in \{1, \ldots, n(x)\}$, be the sequence resulting from deleting the $i$-th token from the sequence $x$, resulting in
$$\text{del}(x, i) = (x^1, \ldots, x^{i-1}, x^{i+1}, \ldots, x^{n(x)}). \tag{11}$$

▶ Let $\text{sub}(x, i, a)$, $x \in \mathcal{X}, i \in \{1, \ldots, n(x)\}, a \in \mathcal{T}$, be the sequence resulting from substituting the token value $a$ into position $i$ of the sequence $x$, resulting in
$$\text{sub}(x, i, a) = (x^1, \ldots, x^{i-1}, a, x^{i+1}, \ldots, x^{n(x)}). \tag{12}$$

| Generated tokens: | $t = 0$ | | $t = 1$ |
|---|---|---|---|

```
def is_prime(n: int) -> bool:
    """Check if a number is prime or not.
    """
    if n < 2: return False
    for i in range(2, n):
        if n % i == 0: return False
    return True
```

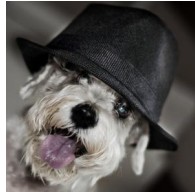

A small white dog wearing a black hat on top of its head.

Figure 4: Edit Flow generation examples with $X_0 = \emptyset$ (i.e. insert-only model). The tokens are color coded to denote the timestep that they were generated in. Left: Coding model conditioned on the function signature. Right: Image captioning model conditioned on the image.

These edit operations define the support of the rate $u_t^\theta(\cdot|x_t)$. Figure 1 shows an example of a CTMC transitioning through sequences using edit operations. Since insertions, deletions, and substitutions result in sequences that are mutually exclusive, we can parameterize each separately.

$$u_t^\theta(\text{ins}(x,i,a)|x) = \lambda_{t,i}^{\text{ins}}(x)Q_{t,i}^{\text{ins}}(a|x) \qquad \text{for } i \in \{1,\ldots,n(x)\} \tag{13}$$

$$u_t^\theta(\text{del}(x,i)|x) = \lambda_{t,i}^{\text{del}}(x) \qquad \text{for } i \in \{1,\ldots,n(x)\} \tag{14}$$

$$u_t^\theta(\text{sub}(x,i,a)|x) = \lambda_{t,i}^{\text{sub}}(x)Q_{t,i}^{\text{sub}}(a|x) \qquad \text{for } i \in \{1,\ldots,n(x)\} \tag{15}$$

With this parameterization, the $\lambda_{t,i} \geq 0$ are the total rates of inserting, deleting, or substituting any token at position $i$ and determines the chances of each operation occurring; $Q_{t,i}^{\text{ins}}(a|x)$ and $Q_{t,i}^{\text{sub}}(a|x)$ are the (normalized) distributions over token values if an insertion or substitution occurs at position $i$. Equations (13)-(15) ensure rates are non-negative and the summation to satisfy (2) is more tractable:

$$u_t^\theta(x_t|x_t) = -\sum_{i=1}^{n(x_t)} \lambda_{t,i}^{\text{ins}}(x_t) - \sum_{i=1}^{n(x_t)} \lambda_{t,i}^{\text{del}}(x_t) - \sum_{i=1}^{n(x_t)} \lambda_{t,i}^{\text{sub}}(x_t). \tag{16}$$

Figure 2 shows the model outputs corresponding to (13)-(15).

**Special cases.** The framework of Edit Flows actually generalizes many existing constructions, as one can restrict the rates to recover existing discrete generative models. For instance, the token-wise probability paths (8) are substitution-only, *i.e.* $\lambda_{t,i}^{\text{ins}} = \lambda_{t,i}^{\text{del}} = 0$, with the mask construction having an additional constraint $\lambda_{t,i}^{\text{sub}}(x) = 0$ if $x^i \neq \text{m}$. As such, the token-wise CTMCs are incapable of increasing or decreasing sequence length. An autoregressive model can also be recovered by only allowing insertions to occur at the rightmost location, *i.e.*, all rates are zero except $\lambda_{t,n(x)}^{\text{ins}}$. As such, the model is incapable of making corrections to the existing sequence other than inserting new tokens in a prescribed order. It can be seen that Edit Flows is a simple yet natural generalization of these existing discrete generative modeling constructions.

## 3.2 Training Edit Flows

Since Edit Flows generalizes beyond the token-wise paths that have been previously explored, it cannot easily make use of existing cross-entropy or evidence lower bound objectives for training, as these are difficult or intractable to derive. The main difficulty in deriving a conditional rate (5) that lies in $\mathcal{X}$ is the need to account for all possible transitions that can transport from one sequence to another, such as multiple possible insertions that transition to the equivalent sequence. Instead, we propose an extension of the DFM training recipe to include an auxiliary Markov process, and in doing so, resulting in allowing Bregman divergences for training Edit Flows.

**Discrete Flow Matching with auxiliary Markov processes.** Suppose we wish to train a CTMC that lies in a space $\mathcal{X}$ and it follows the marginals of a CTMC that lies in an augmented space $(x, z) \in \mathcal{X} \times \mathcal{Z}$ with a probability path $p_t(x, z)$. We show that it is possible to recover the CTMC that transports directly in $\mathcal{X}$, automatically inferring the auxiliary process in $\mathcal{Z}$. This is concisely formalized in the following Theorem 3.1. Further details and proofs are provided in Appendix B. We note that in contrast to the original Flow Matching derivation (Lipman et al., 2024), this result shows that we can marginalize over *time-dependent processes*, not just time-independent variables. Finally, this result is more generally applicable than just training Edit Flows; we showcase another application

of Theorem 3.1 in Appendix C.1 to train with localized propagation rates which incentivizes localized edits, going beyond existing independent probability paths.

**Theorem 3.1** (Flow Matching with Auxiliary Processes). *Let $u_t(x, z|x_t, z_t)$ be a rate over the augmented space of $\mathcal{X} \times \mathcal{Z}$ that generates $p_t(x, z)$, then*

$$u_t(x|x_t) \triangleq \sum_z \mathbb{E}_{p_t(z_t|x_t)} u_t(x, z|x_t, z_t) \qquad generates \qquad p_t(x) \triangleq \sum_z p_t(x, z), \qquad (17)$$

*and furthermore, for any Bregman divergence $D_\phi(a, b) = \phi(a) - \phi(b) - \langle a - b, \frac{\mathrm{d}}{\mathrm{d}b}\phi(b)\rangle$ defined by a convex function $\phi$, we have that*

$$\frac{\mathrm{d}}{\mathrm{d}\theta}\mathbb{E}_{x_t, z_t \sim p_t(x, z)} D_\phi\left(\sum_z u_t(\cdot, z|x_t, z_t), u_t^\theta(\cdot|x_t)\right) = \frac{\mathrm{d}}{\mathrm{d}\theta}\mathbb{E}_{x_t \sim p_t(x)} D_\phi\left(u_t(\cdot|x_t), u_t^\theta(\cdot|x_t)\right). \quad (18)$$

**Training with an auxiliary alignment process.** As previously mentioned, it is difficult to directly construct a conditional rate (5) for Edit Flows, even if given points $x_0$ and $x_1$, as there can be multiple sets of edit operations that transitions from $x_0$ to $x_1$. Instead, we can consider an augmented space where a simpler construction exists. In particular, we will define an auxiliary process using *alignments*.

Given two sequences $x_0$ and $x_1$, an alignment can be used to define a precise set of edit operations that transform $x_0$ to $x_1$. In general, there are many possible alignments for every pair of sequences. For example, below are illustrations of three example alignments between the words 'kitten' and 'smitten' (the most optimal, a sub-optimal padding-to-the-right strategy, and the least optimal):

```
K ε I T T E N        K I T T E N ε ε ε ε ε ε        K I T T E N ε ε ε ε ε ε
↓ ↓                  ↓ ↓ ↓   ↓ ↓ ↓                  ↓ ↓ ↓ ↓ ↓ ↓ ↓ ↓ ↓ ↓ ↓ ↓ ↓
S M I T T E N        S M I T T E N ε ε ε ε ε        ε ε ε ε ε ε S M I T T E N
```

The special token $\varepsilon$ is a *blank* token that *is not added to the vocabulary*, *i.e.*, it is not part of the input or output of the model. Instead, we will only use it to define an auxiliary process that will provide a training signal for Edit Flows via Theorem 3.1. As can be seen, given an alignment, we can recover edit operations as tuples $(a \to b)$ with $a, b \in \mathcal{T} \cup \{\varepsilon\}$, interpreted as an insertion if $a = \varepsilon$, a deletion if $b = \varepsilon$, or a substitution if $a \neq \varepsilon$ and $b \neq \varepsilon$.

Formally, let us define the space of aligned sequences as $\mathcal{Z} = (\mathcal{T} \cup \{\varepsilon\})^N$. Furthermore, we define the function $f_{\text{rm-blanks}} : \mathcal{Z} \to \mathcal{X}$ as the operation of stripping away all the $\varepsilon$ tokens. Note that since this is a many-to-one function, this implies $|\mathcal{X}| < |\mathcal{Z}|$. Following the DFM recipe, we would need to prescribe a coupling $\pi$ and a conditional CTMC that transports from point to point. Given samples from the source $x_0 \sim p(x)$ and target $x_1 \sim q(x)$ in $\mathcal{X}$, we can directly construct aligned sequences $z_0$ and $z_1$ in $\mathcal{Z}$, *e.g.*, by randomly padding the sequences, or by solving for the optimal alignment that corresponds to the minimal edit distance. This defines a coupling $\pi(z_0, z_1)$ over the auxiliary variables satisfying the correct marginal distributions

$$p(x) = \sum_{z_0} \sum_{z_1} \pi(z_0, z_1)\delta_{f_{\text{rm-blanks}}(z_0)}(x), \qquad q(x) = \sum_{z_0} \sum_{z_1} \pi(z_0, z_1)\delta_{f_{\text{rm-blanks}}(z_1)}(x). \quad (19)$$

Then, given $z_0, z_1 \sim \pi$, we define a conditional probability path over the augmented space of $\mathcal{X} \times \mathcal{Z}$

$$p_t(x, z|x_0, z_0, x_1, z_1) = p_t(x, z|z_0, z_1) = p_t(z|z_0, z_1)\delta_{f_{\text{rm-blanks}}(z)}(x), \qquad (20)$$

where $p_t(z|z_0, z_1)$ is a token-wise mixture probability path (8). A conditional rate that transports along the augmented probability path is then given by (see Lemma B.2)

$$u_t(x, z|x_t, z_t, z_0, z_1) = \delta_{f_{\text{rm-blanks}}(z)}(x) \sum_{i=1}^N \frac{\dot{\kappa}_t}{1 - \kappa_t}(\delta_{z_1^i}(z^i) - \delta_{z_t^i}(z^i))\delta_{z_t}(z^{\neg i}) \qquad (21)$$

Note that this rate only transports between sequences $x_t \to x$ that *differ by one edit operation*, perfectly mapping to Edit Flow's transitions (13)-(15). Applying Theorem 3.1, the marginal rate that transports from $p(x)$ to $q(x)$ can be expressed as

$$u_t(x|x_t) = \sum_z \mathbb{E}_{p_t(z_0, z_1, z_t|x_t)} u_t(x, z|x_t, z_t, z_0, z_1), \qquad (22)$$

which we learn using a Bregman divergence as the training loss (see Appendix B.1), simplifying to

$$\mathcal{L}(\theta) = \mathbb{E}_{\substack{\pi(z_0, z_1) \\ t, p_t(x_t, z_t|z_0, z_1)}} \left[\sum_{x \neq x_t} u_t^\theta(x|x_t) - \sum_{i=1}^N \mathbb{1}_{[z_1^i \neq z_t^i]} \frac{\dot{\kappa}_t}{1 - \kappa_t} \log u_t^\theta(x(z_t, i, z_1^i)|x_t)\right] \qquad (23)$$

where $x(z_t, i, z_1^i) = f_{\text{rm-blanks}}(z_t^1, \ldots, z_t^{i-1}, z_1^i, z_t^{i+1}, \ldots, z_t^N)$, which directly corresponds to one of

the edit operations in (13)-(15). This loss can be interpreted as minimizing all the output rates of the model, while having a weighted cross-entropy over edit operations that bring $x_t$ closer to $x_1$.

Interestingly, even when trained with the least optimal alignment, which deletes all tokens from $x_0$ and inserts all tokens in $x_1$, the trained model has a preference towards minimizing the number of edits during its generation process (see Appendix F), learning a non-trivial coupling between $x_0$ and $x_1$. This is analogous to the kinetic energy minimization that is observed for Flow Matching in continuous space (Shaul et al., 2023).

### 3.3 Algorithms and advanced techniques for Edit Flows

In this section, we provide details on the sampling procedure and advanced techniques that make use of the Edit Flows framework. We only provide a summary of each technique here, focusing on the resulting algorithmic procedures and high-level intuition; complete details are in Appendix C.

**Sampling.** Sampling from the model requires transporting a source sample $X_0 \sim p$ to time $t = 1$, simulating the CTMC defined with the learned rate $u_t^\theta$. Following previous works (Campbell et al., 2022; Gat et al., 2024), we leverage the first-order approximation in (1). Sampling thus iterates: with current state $X_t$ and step size $h$, independently determine whether each insertion, deletion and substitution, occurs with probability $h\lambda_{t,i}(X_t)$, then perform all edit operations simultaneously.

**Classifier-free guidance.** We considered a few approaches to add classifier-free guidance (CFG; Ho and Salimans 2022) to Edit Flows. The scheme that we found to be the most reliable, and which we use throughout all experiments, is to apply CFG independently to $\lambda$ and $Q$.

**Sharpening $Q$.** We also explored ad-hoc adjustments to the $Q$ distributions, such as temperature, top-$p$ and top-$k$ sampling, generally intended to sharpen the distribution over the most likely values.

**Reverse rates.** We can also formulate and learn a CTMC that transports from $q$ to $p$. We call this a reverse rate $\breve{u}_t^\theta$ as we apply it in reverse time, from $t = 1$ to $t = 0$. Combining the forward and reverse rates allows us to introduce a stationary component that corrects the samples but does not modify the distribution of the samples, *introducing extra inference-time computation for the ability to self-correct during sampling*. When applied in practice, we take a step forwards in time with $u_t^\theta$ to $t + h(1 + \alpha_t)$ for $\alpha_t > 0$ followed by a step in reverse time with $\breve{u}_{t+h(1+\alpha_t)}^\theta$ back to $t + h$.

**Localized edit operations.** The default rates that we use for the alignments $z_t$ have been factorized per token (21), resulting in independent edit operations. While this allows the use of conditional rates from prior work (8), this could be problematic for Edit Flows as when the sequence length becomes large, noisy sequences $x_t$ will consist of non-neighboring tokens. Instead, we propose a non-factorized locality-based construction in which *if an edit operation has occurred, it incites nearby edit operations to occur*, thereby encouraging locally consistent subsequences in $x_t$. We construct this by creating a novel auxiliary CTMC that locally propagates the occurrence of edit operations in $\mathcal{Z}$ space, and applying Theorem 3.1 to easily obtain a tractable training objective. All details can be found in Appendix C.1. We find localized Edit Flow models to be especially more performant at generating long sequences, leading to a 48% increase in Pass@1 on code generation.

## 4 Related work

**Discrete diffusion and flows for language modeling.** Generative models based on iterative refinement such as diffusion (Sohl-Dickstein et al., 2015; Ho et al., 2020) and flow models (Lipman et al., 2024) have seen their fair share of discrete adaptations. Both aim to learn a CTMC-based generative model but approach the construction differently. Discrete diffusion models typically start with a corruption process which is then reversed (Austin et al., 2021; Lou et al., 2024). Discrete flow models, in contrast, aim to transport between two distributions with an interpolating scheme (Campbell et al., 2024b; Gat et al., 2024). With the DFM framework, Shaul et al. (2024) also proposed new ways of constructing general discrete token-wise paths. However, despite the large design space, none have been able to reliably surpass the simple mask construction, which has been the core focus of many recent works (Sahoo et al., 2024; Shi et al., 2024; Ou et al., 2024; Zheng et al., 2024), motivated by the success of masked language modeling (Devlin et al., 2019; Ghazvininejad et al., 2019; Yang et al., 2019; Chang et al., 2022). In particular, the mask construction has shown to perform well at scale, though it is currently still shy of autoregressive models on code generation tasks and requires heuristic or semi-autoregressive sampling schemes (Nie et al., 2025; Ye et al., 2025; Ermon et al., 2025). In

stark contrast, we explored in the opposite direction, making full use of the CTMC-based construction instead of simplifying it. This allowed us to generalize the existing DFM construction to enable variable-length generation and construct a model using position-relative edits as a generative process.

**Non-autoregressive variable length generation.** When the generative modeling framework does not inherently allow variable length generation, such as many non-autoregressive approaches, the stereotypical method of handling it is to utilize a separate length prediction model (*e.g.* Lee et al. 2018). More integrated approaches have considered edit operations, though many of the existing constructions are heuristic-based and do not show that they properly sample from the target distribution. Levenshtein Transformer (Gu et al., 2019b) and DiffusER (Reid et al., 2022) are edit-based sequence generation models. They consider a sequential expert policy that performs a series of edits at each step, and the model is trained through imitation learning. Unlike Edit Flows, DiffusER uses a causal masked model (Aghajanyan et al., 2022) to fill in insertions and substitutions autoregressively and is trained to match a discrete-time corruption process that is sequentially simulated. Chan et al. (2020) considers sequence alignments using only deletion operations and leverages marginalization over latent alignments. Gu et al. (2019a) and Stern et al. (2019) propose insertion-only models that sequentially predict what and where to insert tokens. The most similar work to ours is perhaps Campbell et al. (2024a), who proposed modeling inserts in a jump diffusion framework, relying on generator theory and evidence lower bounds for training. However, extending this direct derivation approach to more than a singular insertion, and to introduce deletions and substitutions, is very challenging and arguably intractable; an issue that we got around by making simple use of Theorem 3.1.

**Relative positions for language modeling.** There is a growing trend to incorporate only relative positional information into neural network architectures (Liutkus et al., 2021; Press et al., 2021; Peebles and Xie, 2023; Su et al., 2024; Ding et al., 2024). However, on the methods side, there has not yet been a shift due to non-autoregressive models mainly using a token-wise construction. As such, every token generated must also account for the exact position (*e.g.*, exact number of neighboring mask tokens) when deciding on a token value. Edit Flows is one of the first models to use only relative and localized operations in the method construction, sample generation time, and in the architecture. Beyond the capability of variable length generation, enabling the use of position-relative generation may be a key advancement and could be the underlying reason that allows Edit Flows to outperform methods based on absolute positioning.

**Iterative editing models.** Several prior works on constrained generation employ iterative editing or sampling procedures. Welleck et al. decouple an existing language generator from a learned iterative corrector that refines its outputs, whereas Edit Flows uses a single model to begin from a random or null sequence and directly generate outputs through a sequence of discrete token edits. Miao et al. (2019) employ Metropolis–Hastings sampling over insertion/deletion/replacement operations to satisfy lexical constraints; by contrast, Edit Flows deterministically takes a fixed number of flow-matching steps from noise to data without an acceptance criterion. Qin et al. (2022) propose COLD decoding, an energy-based approach that iteratively refines whole sequences via Langevin dynamics under constraints, whereas Edit Flows incrementally edits tokens rather than resampling full sequences. Finally, Sha (2020) formulates lexically-constrained generation as a gradient-guided optimization problem using a differentiable fluency objective to guide edits, but Edit Flows requires no external objective or backpropagation at test time. These contrasts underscore that Edit Flows integrates generation and editing within a single flow-matching model, rather than relying on separate generation and correction modules or auxiliary sampling schemes.

# 5 Experiments

We experimentally validate the performance of Edit Flows on multiple text generation tasks, including image-to-text generation using 280M models, text and code generation benchmarks with 1.3B models.

*Baselines.* We primarily compare against a state-of-the-art **Autoregressive** model (Vaswani et al., 2017; Touvron et al., 2023) with standard left-to-right generation, and **Mask DFM** (Gat et al., 2024) which is the most relevant and best performing non-autoregressive framework currently for text generation, equivalent to discrete mask diffusion models.

*Models.* We test two variants of our models with different $p(X_0)$. For the default **Edit Flow** we use $p = \delta_\emptyset$ so that the flow generates using a combination of insertions and deletions, with the forward and reverse rates, respectively. A variant **Uniform** $X_0$ **+ Edit Flow** use $X_0 = (X^1, X^2, \ldots, X^{100})$ where $X^i \sim p_{\text{emp}}$, with $p_{\text{emp}}$ being the (marginalized) empirical dis-

| Method | MS COCO | | | Image Captioning 3M | | |
|---|---|---|---|---|---|---|
| | METEOR | CIDEr | SPICE | ROUGE-L | CIDEr | SPICE |
| VLP[§] (Zhou et al., 2020) | 28.4 | 117.7 | 21.3 | 24.3 | 77.5 | 16.5 |
| ClipCap[§] (Mokady et al., 2021) | 27.1 | 108.3 | 20.1 | 26.7 | 87.2 | 18.5 |
| Llama3 Autoregressive | 25.7 | 95.5 | 19.6 | 25.2 | 85.8 | 17.8 |
| Mask DFM | 25.3 | 95.6 | 19.2 | 27.4 | 96.2 | 20.3 |
| Edit Flow (**Ours**) | 27.4 | 108.1 | 21.1 | 29.0 | 101.9 | 21.7 |
| Localized Edit Flow (**Ours**) | 27.4 | 105.1 | 22.1 | 28.3 | 99.7 | 20.8 |

Table 1: Image captioning benchmarks using Llama3 280M models. [§]These works used pretrained models that were trained on larger amount of data and cannot be directly compared; they are shown for reference only. Colors show the best and second best among each metric.

| Method | HellaSwag | ARC-E | ARC-C | PIQA | OBQA | WinoGrande |
|---|---|---|---|---|---|---|
| Llama3 Autoregressive | 49.5 | 71.0 | 36.3 | 76.0 | 30.4 | 62.1 |
| Mask DFM | 38.3 | 55.4 | 27.8 | 65.3 | 22.6 | 52.3 |
| Edit Flow (**Ours**) | 49.0 | 63.1 | 33.0 | 68.8 | 28.6 | 53.6 |

Table 2: Zero-shot text benchmarks using Llama3 1.3B parameter models trained on DCLM-baseline 1.0 (Li et al., 2024).

tribution of the tokens in the training set. When constructing the alignment between $z_0$ and $z_1$, 50 of the initial tokens are deleted and the other 50 are substituted, with the remaining tokens inserted. Finally, a **Localized Edit Flow** that makes use of a localized propagation process Appendix C.1, which encourages localized edits during generation.

*Architecture and hyperparameters.* We use 280M and 1.3B parameter variants of the Llama architecture (Grattafiori et al., 2024; Touvron et al., 2023) for all of our models and baselines. The maximum sequence length during training is set to 1024 tokens for all models. The Autogressive baseline uses causal attention, while the Mask DFM and Edit Flow models use full self-attention, including an additional token encoding the value of $t$. For Edit Flow, we use FlexAttention (Dong et al., 2024) to handle batches of variable lengths, allowing us to not require special padding tokens and significantly increasing token efficiency during training. In our experiments, Edit Flows are able ingest $3\times$ more training data per iteration while using the same compute and memory as Mask DFM. We train all models and baselines using the same compute budget for fair comparison. We use a cubic scheduler $\kappa_t = t^3$ for Edit Flows and Mask DFM, which we found to perform better than the linear scheduler as also observed by Gat et al. (2024). Further hyperparameter details are in Appendix E.

| | Method | HumanEval | | HumanEval+ | | MBPP | |
|---|---|---|---|---|---|---|---|
| | | Pass@1 | Pass@10 | Pass@1 | Pass@10 | Pass@1 | Pass@10 |
| | Autoregressive (Gat et al., 2024) | 14.3 | 21.3 | | | 17.0 | 34.3 |
| | Llama3 Autoregressive[†] | 17.0 | 34.7 | 14.0 | 28.6 | 25.6 | 45.4 |
| Non-AR | Mask DFM (Gat et al., 2024) | 6.7 | 13.4 | | | 6.7 | 20.6 |
| | Mask DFM (Oracle Length) (Gat et al., 2024) | 11.6 | 18.3 | | | 13.1 | 28.4 |
| | Mask DFM[†] | 9.1 | 17.6 | 7.9 | 13.4 | 6.2 | 25.0 |
| | Uniform $X_0$ + Edit Flow (**Ours**) | 9.7 | 24.3 | 9.7 | 19.5 | 9.4 | 33.4 |
| | Edit Flow (**Ours**) | 12.8 | 24.3 | 10.4 | 20.7 | 10.0 | 36.4 |
| | Localized Edit Flow (**Ours**) | 14.0 | 22.6 | 10.4 | 18.9 | 14.8 | 34.0 |

Table 3: Code generation benchmarks using Llama3 1.3B parameter models trained on the CodeLlama (Roziere et al., 2023) datamix. [†]Superscript denotes our own implementation. We highlight the best non-autoregressive models, where colors show the best and second best among each metric.

**Image captioning.** We train on the task of image to text generation, using image captioning datasets for training and validation. Specifically, we train from scratch on the MS COCO dataset (Lin et al. 2014; CC-BY 4.0) and an image captioning dataset containing 3M image-caption pairs. Results are shown in Table 1, where we also provide prior works as references that used large pretrained models. By training on the larger Image Captioning 3M dataset, our models can match the performance of these references. We see that for generation of short sequences such as captions, non-autoregressive models can be better than autoregressive models. Furthermore, we see a sizeable improvement in performance from using our Edit Flow models. We attribute this improvement to the native capabilities of handling variable lengths. We see that the Localized Edit Flow performs on par but does not outperform the default Edit Flow, which is expected for short length generation. Examples of the generation process are shown in Figure 5.

**Text benchmarks.** For text benchmarks, we trained our models using the DCLM baseline 1.0 (Li et al. 2024; CC-BY 4.0) dataset. We show the results for common text benchmarks in Table 2. Following (Nie et al., 2025), we perform CFG during evaluation, which has multiple ways to be extended when applied to general CTMC processes. The Edit Flow model is significantly better than the Mask DFM model, but it is slightly behind Autoregressive.

**Code benchmarks.** For the code generation benchmarks, we used the CodeLlama datamix (Roziere et al., 2023). Results are shown in Table 3. As additional baselines, we compare against the results reported by Gat et al. (2024), which includes an oracle where the ground truth length is provided to the model. Interestingly, we see that Edit Flows can outperform even the model with oracle length provided. We note that on such large scale data sets, the lengths of the sequence seen during training are not very informative and we need to crop sequences to a maximum length anyhow (see Figures 9,10); however, the ability of Edit Flows to generate and process using only relative positions still gives Edit Flow a superior edge. Furthermore, our Edit Flow models are competitive with the Autoregressive model reported by Gat et al. (2024), though it still falls short compared to our own implementation. An interesting result is that the Localized Edit Flow model significantly outperforms the other non-autoregressive models on MBPP, which is known to require generating long sequences of code, with a *relative improvement of 48% at Pass@1* over the non-localized Edit Flow and a 138% relative improvement over Mask DFM.

# 6  Limitations

We identify two key limitations in our empirical results.

The pre-training configuration we employed favors autoregressive models. During autoregressive training, the model is exposed to all possible conditioning contexts within the input sequence. In contrast, we selected a random subset of the input to serve as conditioning. If this randomly chosen subset does not closely align with the evaluation scenario, the resulting learning signal may be diminished.

The text benchmarks used in our study focus on likelihood estimation rather than text generation. Since non-autoregressive models lack a closed-form expression for likelihood, we relied on the best available alternative—a noisy estimate of the ELBO. Although this provides a reasonable approximation, these benchmarks do not directly assess the quality of the generated text produced by the models.

# 7  Conclusion

Edit Flows operate using position-relative edit operations and naturally support variable-length generation. By modeling sequence generation as a CTMC, our approach captures expressive sequence-level transition dynamics without relying on rigid, factorized processes. Empirically, Edit Flows show consistent improvement over the mask construction across a range of large scale benchmarks. In our initial results, they surpass autoregressive models in image captioning and but fall slightly behind them in text-benchmarks and code generation. However, many training pipelines and benchmarks are designed for autoregressive models, and we believe that further efforts can significantly boost performance.

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

# A Qualitative Examples

| Input Image | Edit Flow caption generation process |
| --- | --- |
|  | dog
dog.
dog television.
dog a television.
doges an animal a television.
A doges of an animal a television.
A brown and doges of an animal a television.
A brown and dog watches of an animal a television.
A brown and white dog watches of an on a television.
A brown and white dog watches an image of an animal on a television. |
|  | black
black hat
aring black hat
dogaring black hat head.
dogaring black hat on head.
A dogaring black hat on top head.
A dog wearing black hat on top head.
A white dog wearing black hat on top of head.
A white dog wearing a black hat on top of head.
A small white dog wearing a black hat on top of its head. |
|  | a
a tree
a tree a
a a tree a
a of a tree a
a close of on a tree a
a close up of birds on a tree branch a
a close up of birds on a tree branch a pot
a close up of birds on a tree branch with a pot |
|  | over
over.
lies over a.
lies over a street.
Anlies over a street.
Anlies over a street cars.
Anlies over street with cars.
An flies over street with cars.
An air flies over street with cars.
An airplane flies over street with cars.
An airplane flies over a street with cars. |

Figure 5: Example input images and the stochastic sequential generation of captions from an Edit Flows model.

Generated tokens:    $t = 0$ ██████████████████████████████ $t = 1$

```python
def truncate_number(number: float) -> float:
    """ Given a positive floating point number, it can be decomposed into
    and integer part (largest integer smaller than given number) and decimals
    (leftover part always smaller than 1).

    Return the decimal part of the number.
    >>> truncate_number(3.5)
    0.5
    """
    return number - int(number - 0.0)
```

```python
from typing import List, Tuple

def sum_product(numbers: List[int]) -> Tuple[int, int]:
    """ For a given list of integers, return a tuple consisting of a sum and a product of all the integers in a list.
    Empty sum should be equal to 0 and empty product should be equal to 1.
    >>> sum_product([])
    (0, 1)
    >>> sum_product([1, 2, 3, 4])
    (10, 24)
    """
    sum = 0
    product = 1

    if not numbers:
        return sum, product

    for index, number in enumerate(numbers):
        sum += index
        product *= number
    return sum, product

def sum_product_empty(numbers: List[int]) -> Tuple[int, int]:
    """
    >>> sum_product([])
    (0, 1)
    >>> sum_product([1, 2, 3, 4])
    (10, 24)
    """

if __name__ == '__main__':
    print(sum_product())
    print(sum_product_empty([6]))
```

```python
def string_sequence(n: int) -> str:
    """ Return a string containing space-delimited numbers starting from 0 upto n inclusive.
    >>> string_sequence(0)
    '0'
    >>> string_sequence(5)
    '0 1 2 3 4 5'
    """
    numbers = []
    # add numbers to sequence
    numbers += [str(value) for value in range(n+1)]
    # space delimited
    return str(' '.join(numbers))
```

Figure 6: Edit Flow generation examples with $X_0 = \emptyset$ (i.e. insert-only model) without a divergence-free component. 100 sampling steps. The function signature and the docstring serve as prompts.

**Initial State (Step 0):**

```python
def is_prime(n: int) -> bool:
    if n <= 1:
        return True
    for i in range(2, n):
        if i % n == 0:
            return True
    return False
```

**Intermediate State (Step 150):**

```python
def is_prime(n: int) -> bool:
    if n <= 1:
        return False
    for i in range(2, n):
        if . % n == 0:
            return False
  True

,
6
```

**Final State (Step 300):**

```python
def is_prime(n: int) -> bool:
    if n <= 1:
        return False
    for i in range(2, n):
        if n % i == 0:
            return False
    return True
```

Figure 7: Edit Flow error correction example for correcting the `is_prime` function. The initial implementation is incorrect, because the three return statement are negated. The model starts from an incorrect implementation, makes 117 edits over 300 steps, and reaches the correct final state. Note that intermediate states may contain extra tokens that are later deleted.

# B   Theorems and proofs

**Theorem 3.1** (Flow Matching with Auxiliary Processes). *Let $u_t(x, z|x_t, z_t)$ be a rate over the augmented space of $\mathcal{X} \times \mathcal{Z}$ that generates $p_t(x, z)$, then*

$$u_t(x|x_t) \triangleq \sum_z \mathbb{E}_{p_t(z_t|x_t)} u_t(x, z|x_t, z_t) \qquad generates \qquad p_t(x) \triangleq \sum_z p_t(x, z), \qquad (17)$$

*and furthermore, for any Bregman divergence $D_\phi(a, b) = \phi(a) - \phi(b) - \langle a - b, \frac{d}{db}\phi(b)\rangle$ defined by a convex function $\phi$, we have that*

$$\frac{d}{d\theta}\mathbb{E}_{x_t, z_t \sim p_t(x, z)}D_\phi\left(\sum_z u_t(\cdot, z|x_t, z_t), u_t^\theta(\cdot|x_t)\right) = \frac{d}{d\theta}\mathbb{E}_{x_t \sim p_t(x)}D_\phi\left(u_t(\cdot|x_t), u_t^\theta(\cdot|x_t)\right). \quad (18)$$

*Proof.* For the first part of the theorem (17), since $u_t(x, z|x_t, z_t)$ generates $p_t(x, z)$, they satisfy the Kolmogorov forward equation

$$\frac{\partial}{\partial t}p_t(x, z) = \sum_{x_t}\sum_{z_t} p_t(x_t, z_t)u_t(x, z|x_t, z_t),$$

then we can show $u_t(x|x_t)$ and $p_t(x)$ also satisfy the Kolmogorov forward equation

$$\frac{\partial}{\partial t} p_t(x) = \sum_z \frac{\partial}{\partial t} p_t(x,z) = \sum_z \sum_{x_t} \sum_{z_t} p_t(x_t, z_t) u_t(x, z|x_t, z_t)$$

$$= \sum_{x_t} \underbrace{\sum_z \sum_{z_t} u_t(x, z|x_t, z_t) \frac{p_t(x_t, z_t)}{p_t(x_t)}}_{u_t(x|x_t)} p_t(x_t)$$

$$= \sum_{x_t} p_t(x_t) u_t(x|x_t).$$

Additionally, $u_t(x, z|x_t, z_t)$ satisfies the rate conditions by assumption. Assume $p_t(x_t) > 0$. Then $\sum_x u_t(x|x_t) = \sum_{z_t} (\sum_x \sum_z u_t(x, z|x_t, z_t) 1(p_t(x_t, z_t) > 0)) \frac{p_t(x_t, z_t)}{p_t(x_t)} = 0$. Further, $u_t(x|x_t) \geq 0$ when $x \neq x_t$ and $p_t(x_t) > 0$ because $u_t(x, z|x_t, z_t) \geq 0$ when $(x, z) \neq (x_t, z_t)$ and $p_t(x_t, z_t) > 0$. Terms with $p_t(x_t, z_t) = 0$ do not contribute in the sum. So $u_t(x|x_t)$ satisfies the rate conditions.

For the second part of the theorem (18), note that

$$\mathbb{E}_{x_t, z_t \sim p_t(x,z)} \sum_x \sum_z u_t(x, z|x_t, z_t) \frac{\mathrm{d}}{\mathrm{d}u} \phi(u_t^\theta(x|x_t))$$

$$= \sum_{x_t} \sum_{z_t} \sum_x \sum_z \frac{p_t(x_t, z_t)}{p_t(x_t)} p_t(x_t) u_t(x, z|x_t, z_t) \frac{\mathrm{d}}{\mathrm{d}u} \phi(u_t^\theta(x|x_t))$$

$$= \sum_{x_t} \sum_x p_t(x_t) u_t(x|x_t) \frac{\mathrm{d}}{\mathrm{d}u} \phi(u_t^\theta(x|x_t))$$

$$= \mathbb{E}_{x_t \sim p_t(x)} \sum_x u_t(x|x_t) \frac{\mathrm{d}}{\mathrm{d}u} \phi(u_t^\theta(x|x_t))$$

then we can directly prove the result

$$\frac{\mathrm{d}}{\mathrm{d}\theta} \mathbb{E}_{x_t, z_t \sim p_t(x,z)} D_\phi \left( \sum_z u_t(\cdot, z|x_t, z_t), u_t^\theta(\cdot|x_t) \right)$$

$$= \frac{\mathrm{d}}{\mathrm{d}\theta} \mathbb{E}_{x_t, z_t \sim p_t(x,z)} \left[ \phi(\sum_z u_t(\cdot, z|x_t, z_t)) - \phi(u_t^\theta(\cdot|x_t)) - \langle \sum_z u_t(\cdot, z|x_t, z_t) - u_t^\theta(\cdot|x_t), \frac{\mathrm{d}}{\mathrm{d}u} \phi(u_t^\theta(\cdot|x_t)) \rangle \right]$$

$$= \frac{\mathrm{d}}{\mathrm{d}\theta} \mathbb{E}_{x_t, z_t \sim p_t(x,z)} \left[ -\phi(u_t^\theta(\cdot|x_t)) - \sum_x \sum_z u_t(x, z|x_t, z_t) \frac{\mathrm{d}}{\mathrm{d}u} \phi(u_t^\theta(x|x_t)) + \sum_x u_t^\theta(x|x_t) \frac{\mathrm{d}}{\mathrm{d}u} \phi(u_t^\theta(x|x_t)) \right]$$

$$= \frac{\mathrm{d}}{\mathrm{d}\theta} \mathbb{E}_{x_t \sim p_t(x)} \left[ \phi(u_t(\cdot|x_t)) - \phi(u_t^\theta(\cdot|x_t)) - \sum_x u_t(x|x_t) \frac{\mathrm{d}}{\mathrm{d}u} \phi(u_t^\theta(x|x_t)) + \sum_x u_t^\theta(x|x_t) \frac{\mathrm{d}}{\mathrm{d}u} \phi(u_t^\theta(x|x_t)) \right]$$

$$= \frac{\mathrm{d}}{\mathrm{d}\theta} \mathbb{E}_{x_t \sim p_t(x)} \left[ \phi(u_t(\cdot|x_t)) - \phi(u_t^\theta(\cdot|x_t)) - \langle u_t(x|x_t) - u_t^\theta(x|x_t), \frac{\mathrm{d}}{\mathrm{d}u} \phi(u_t^\theta(x|x_t)) \rangle \right]$$

$$= \frac{\mathrm{d}}{\mathrm{d}\theta} \mathbb{E}_{x_t \sim p_t(x)} D_\phi(u_t(\cdot|x_t), u_t^\theta(\cdot|x_t))$$

$\square$

To apply theorem 3.1, we require a rate $u_t(x, z|x_t, z_t)$ in the augmented space of $\mathcal{X} \times \mathcal{Z}$ that generates $p_t(x, z)$. The following lemma can simplify this construction.

**Lemma B.1** (Rates that generate $p_t(x, z) = p(x|z) p_t(z)$)**.** *Let $p_t(x, z)$ be a distribution over augmented space of $\mathcal{X} \times \mathcal{Z}$ where $p_t(x|z) = p(x|z)$ is time-independent. Let $u_t(z|z_t)$ be a rate over $\mathcal{Z}$ that generates $p_t(z)$. Then*

$$u_t(x, z|x_t, z_t) = (1 - \delta_{z_t}(z)) p(x|z) u_t(z|z_t) + \delta_{x_t}(x) \delta_{z_t}(z) u_t(z, z_t) \tag{24}$$

*is a rate over augmented space of $\mathcal{X} \times \mathcal{Z}$ that generates $p_t(x, z)$.*

*Proof.* We first check rate conditions (2) for $u_t(x, z|x_t, z_t)$. When $(x, z) \neq (x_t, z_t)$ and $p_t(x_t, z_t) > 0$, $u_t(x, z|x_t, z_t) = (1 - \delta_{z_t}(z))p(x|z)u_t(z|z_t) \geq 0$ because $p_t(z_t) > 0$. Then

$$\sum_{x,z} u_t(x, z|x_t, z_t) = \sum_{x,z} (1 - \delta_{z_t}(z))p(x|z)u_t(z|z_t) + \delta_{x_t}(x)\delta_{z_t}(z)u_t(z, z_t)$$

$$= \sum_z u_t(z|z_t) - u_t(z_t|z_t) + u_t(z_t|z_t)$$

$$= \sum_z u_t(z|z_t)$$

$$= 0.$$

where the last equality uses that $u_t(z|z_t)$ is a rate over $\mathcal{Z}$ and again $p_t(x_t, z_t) > 0 \implies p_t(z_t) > 0$.

Now we show $u_t(x, z|x, z_t)$ also satisfies the Kolmogorov forward equation (3) for $p_t(x, z)$ which proves the result

$$\sum_{x_t, z_t} u_t(x, z|x_t, z_t)p_t(x_t, z_t)$$

$$= \sum_{x_t, z_t} \left( (1 - \delta_{z_t}(z))p(x|z)u_t(z|z_t) + \delta_{x_t}(x)\delta_{z_t}(z)u_t(z, z_t) \right) p_t(x_t, z_t)$$

$$= p(x|z) \sum_{z_t} u_t(z|z_t)p_t(z_t) - u_t(z|z)p_t(x, z) + u_t(z|z)p_t(x, z)$$

$$= p(x|z)\frac{\partial}{\partial t}p_t(z)$$

$$= \frac{\partial}{\partial t}p_t(x, z).$$

$\square$

When the relationship between $x$ given auxiliary $z$ is not only time-independent, but also deterministic, this Lemma B.1 leads to the following Lemma stated inline in the main text

**Lemma B.2** (Rates that generate $p_t(x, z) = \delta_{f(z)}(x)p_t(z)$). *Let* $p_t(x, z) = \delta_{f(z)}(x)p_t(z)$ *be a distribution over augmented space of* $\mathcal{X} \times \mathcal{Z}$ *where* $p_t(x|z) = \delta_{f(z)}(x)$ *is time-independent and deterministic. Let* $u_t(z|z_t)$ *be a rate over* $\mathcal{Z}$ *that generates* $p_t(z)$. *Then*

$$u_t(x, z|x_t, z_t) = \delta_{f(z)}(x)u_t(z|z_t) \tag{25}$$

*is a rate over augmented space of* $\mathcal{X} \times \mathcal{Z}$ *that generates* $p_t(x, z)$.

*Proof.* From Lemma B.1, the rate in equation (24) generates this $p_t(x, z)$ using $u_t(z|z_t)$. Because we only use this rate when $p_t(x_t, z_t) > 0$, this rate will always be evaluated at $x_t = f(z_t)$ giving

$$u_t(x, z, f(z_t), z_t) = (1 - \delta_{z_t}(z))p(x|z)u_t(z|z_t) + \delta_{f(z_t)}(x)\delta_{z_t}(z)u_t(z, z_t)$$

$$= \delta_{f(z)}(x)u_t(z|z_t) + \delta_{z_t}(z)\left(-\delta_{f(z)}(x) + \delta_{f(z_t)}(x)\right)u_t(z|z_t)$$

$$= \delta_{f(z)}(x)u_t(z|z_t).$$

$\square$

## B.1 A Bregman divergence as the training loss for Edit Flows

Given velocities $u_t(\cdot, z|x_t, z_t)$ and $u_t^\theta(\cdot|x_t)$ that satisfy the rate conditions, we define

$$\phi(u_t(\cdot|x_t)) = \sum_{x \neq x_t} u_t(x|x_t) \log u_t(x|x_t). \tag{26}$$

The Bregman divergence corresponding to this $\phi$ is:

$$D_\phi(f(\cdot|x_t), g(\cdot|x_t)) = \phi(f(\cdot|x_t)) - \phi(g(\cdot|x_t)) - \sum_{x \neq x_t} (f(x|x_t) - g(x|x_t))(1 + \ln g(x|x_t))$$

$$= \sum_{x \neq x_t} \left( -f(x|x_t) - f(x|x_t)\ln\frac{g(x|x_t)}{f(x|x_t)} + g(x|x_t) \right) \tag{27}$$

Therefore the training loss for Edit Flows with this $\phi$ can be written

$$\mathcal{L}(\theta) = \mathbb{E}_{t,\pi(z_0,z_1),p_t(x_t,z_t|z_0,z_1)} D_\phi \left( \sum_z u_t(\cdot,z|x_t,z_t), u_t^\theta(\cdot|x_t) \right)$$

$$= \mathbb{E}_{t,\pi(z_0,z_1),p_t(x_t,z_t|z_0,z_1)} \left[ - \sum_{z,x\neq x_t} u_t(x,z|x_t,z_t,z_0,z_1) + \sum_{x\neq x_t} u_t^\theta(x|x_t) \right.$$

$$\left. - \sum_{z,x\neq x_t} u_t(x,z|x_t,z_t,z_0,z_1) \log \frac{u_t^\theta(x|x_t)}{u_t(x,z|x_t,z_t,z_0,z_1)} \right]$$

$$= -\mathbb{E}_{t,\pi(z_0,z_1),p_t(x_t,z_t|z_0,z_1)} \left[ u_t^\theta(x_t|x_t) + \sum_z \sum_{x\neq x_t} u_t(x,z|x_t,z_t,z_0,z_1) \log u_t^\theta(x|x_t) \right] + \text{const.}$$

$$\tag{28}$$

## C  Advanced techniques for Edit Flows

**Sampling.**   Sampling from the model requires transporting a source sample $X_0 \sim p$ to time $t = 1$, simulating the CTMC defined with the learned rate $u_t^\theta$. Exact simulation (Gillespie, 1976, 1977) is intractable as it requires integration of $u_t^\theta$. With the Edit Flow parameterization (13)-(15), the exact probability of an edit operation characterized by the rate $\lambda_{t,i}$ occurring within an interval $(t, t + h)$ is

$$1 - e^{-\int_t^{t+h} \lambda_{t,i}(X_t)\mathrm{d}t} \approx h\lambda_{t,i}(X_t). \tag{29}$$

Following previous works (Campbell et al., 2022; Gat et al., 2024), we leverage the first-order approximation. Sampling thus iterates the following procedure: with current state $X_t$ and step size $h$, independently determine the probability of each insertion, deletion and substitution, then perform all edit operations simultaneously.

1. For each position $i$, sample whether to insert with probability $h\lambda_{t,i}^{\text{ins}}(X_t)$ and whether to delete or substitute with probability $h(\lambda_{t,i}^{\text{ins}}(X_t) + \lambda_{t,i}^{\text{del}}(X_t))$. Since deletions and substitutions at the same position are exclusive, if either occurs, select deletion with probability $\lambda_{t,i}^{\text{del}}(X_t)/(\lambda_{t,i}^{\text{del}}(X_t) + \lambda_{t,i}^{\text{sub}}(X_t))$, otherwise substitution.

2. If insertion or substitution at $i$, sample the new token value from $Q_{t,i}^{\text{ins}/\text{sub}}(\cdot|X_t)$.

3. $t \leftarrow t + h$

**Classifier-free guidance.**   We considered three approaches to add classifier-free guidance to Edit Flows. Classifier-free guidance (CFG) considers training a model with and without conditioning $c$ and combining those two models at sampling time using a weighting hyperparameter $w$.

Our first approach is *weighted rate* CFG which follows Nisonoff et al. 2024 and uses (for $x \neq x_t$ and within one edit operation)

$$\tilde{u}_t(x|x_t,c) \triangleq u_t(x|x_t)^{1-w} u_t(x|x_t,c)^w = \hat{\lambda}_{t,i}(x_t,c)\tilde{Q}_{t,i}(a|x_t,c) \tag{30}$$

$$\text{with} \quad \begin{aligned} \hat{\lambda}_{t,i}(x_t,c) &= \lambda_{t,i}(x_t)^{1-w}\lambda_{t,i}(x_t|c)^w \sum_a Q_{t,i}(a|x_t)^{1-w}Q_{t,i}(a|x_t,c)^w \\ \tilde{Q}_{t,i}(a|x_t,c) &\propto Q_{t,i}(a|x_t)^{1-w}Q_{t,i}(a|x_t,c)^w \end{aligned} \tag{31}$$

where $\lambda_{t,i}$ and $Q_{t,i}$ are for the specific edit operation taking $x_t \to x$.

Our second *fixed rate* CFG which uses $\tilde{u}_t(x|x_t,c) \triangleq \lambda_t(x_t,c)\tilde{Q}_t(a|x_t,c)$.

Our third approach is *naïve rate* CFG which uses $\tilde{u}_t(x|x_t,c) \triangleq \tilde{\lambda}_{t,i}(x_t,c)\tilde{Q}_t(a|x_t,c)$ where $\tilde{\lambda}_t(x_t,c) = \lambda_{t,i}(x_t|c)^{1+w}\lambda_{t,i}(x_t)^{-w}$.

Note that these CFG methods only differ in how the modified $\lambda_{t,i}$ is constructed, impacting the probability of an edit operation. For all of our benchmarks, the *naïve rate* CFG consistently performed the best, with *fixed rate* CFG very close in performance; however, the *weighted rate* CFG was consistently worse than either options. When CFG is applied in conjunction with reverse rates, we applied CFG to both the forward and reverse rates.

**Reverse rates.** A CTMC Markov process can also be defined via reverse time simulation from $t = 1$ to $t = 0$ using rates $\bar{u}_t$

$$\mathbb{P}(X_{t-h} = x | X_t = x_t) = \delta_{x_t}(x) + h\bar{u}_t(x|x_t) + o(h) \tag{32}$$

where $o(h)$ satisfies $\lim_{h\to 0} \frac{o(h)}{h} = 0$. This equation is identical to forward-time simulation (1) except that the transition is from $t$ to $t - h$ instead of $t$ to $t + h$. Like (1), in order for (32) to define a valid probability distribution, reverse rates $\bar{u}_t$ must obey the rate conditions in (2).

A rate $\bar{u}_t$ "generates" a probability path $p_t$ if the time marginals of the associated reverse-time simulation are samples from $p_t$, *i.e.*, $X_t \sim p_t$. Concretely, they should satisfy the Kolmogorov forward equation in reverse (*i.e.*, with a minus sign)

$$-\frac{\partial}{\partial t}p_t(x) = \sum_y \bar{u}_t(x|y)p_t(y) = \underbrace{\sum_{y\neq x} \bar{u}_t(x|y)p_t(y)}_{\text{reverse flow into } x} - \underbrace{\sum_{y\neq x} \bar{u}_t(y|x)p_t(x)}_{\text{reverse flow out of } x}. \tag{33}$$

We can construct $\bar{u}_t$ that generates $p_t$ (and in fact is a CTMC with the same joint distribution) from $u_t$ that generates $p_t$ via the following procedure. Assume $u_t$ generates $p_t$. For $x \neq x'$, consider that the probability flux from $x$ in forward time towards $x'$ equals the probability flux from $x'$ to $x$ in reverse time as follows

$$\underbrace{\bar{u}_t(x|x')p_t(x')}_{\text{reverse flux from } x' \text{ into } x} = \underbrace{u_t(x'|x)p_t(x)}_{\text{flux from } x \text{ into } x'}. \tag{34}$$

Inserting into the Kolmogorov forward equation satisfied by $u_t$

$$\begin{aligned}
\frac{\partial}{\partial t}p_t(x) &= \sum_{y\neq x} u_t(x|y)p_t(y) - \sum_{y\neq x} u_t(y|x)p_t(x) \\
&= \sum_{y\neq x} \bar{u}_t(y|x)p_t(x) - \sum_{y\neq x} \bar{u}_t(x|y)p_t(y) \\
&= -\sum_y \bar{u}_t(x|y)p_t(y),
\end{aligned} \tag{35}$$

so $\bar{u}_t$ generates $p_t$.

Now consider $u_t + \bar{u}_t$, which satisfies (2) and is *probability-preserving* such that $\sum_{x_t}(u_t(x|x_t) + \bar{u}_t(x|x_t))p_t(x_t) = 0$. If we perform forward simulation with this rate using (1) starting from $x \sim p_t(x)$ and sampling $x'$, we maintain that $x' \sim p_t(x)$. This allows *corrector* steps that can correct errors in the marginal distribution via repeatedly applying such a step without updating time.

We also have that $(1+\alpha)u_t + \alpha\bar{u}_t$ for $\alpha \geq 0$ generates $p_t$ in forward time. This combination rate can be simulated via stepping forward from $x_t$ to $x_{t+h(1+\alpha)}$ using $u_t$ and then backwards to $x_{t+h}$ using $\bar{u}_{t+h(1+\alpha)}$. To see this is equivalent for small $h$, let $y = x_{t+h(1+\alpha)}$ and consider the distribution of $x_{t+h}$ after the combination of these two steps

$$\sum_y \left(\delta_y(x_{t+h}) + h\alpha\bar{u}_{t+h(1+\alpha)}(x_{t+h}|y) + o(h)\right)\left(\delta_y(x) + h(1+\alpha)u_t(y|x_t) + o(h)\right)$$

$$= \delta_{x_t}(x_{t+h}) + h\left(\alpha\bar{u}_t(x_{t+h}|x_t) + (1+\alpha)u_t(x_{t+h}|x_t)\right) + o(h). \tag{36}$$

## C.1 Localized propagation paths

Edit Flows leverage an underlying conditional probability path $p_t(z|z_0, z_1)$ and associated rates $u_t(z|z_t)$, so far given by the factorized token-wise mixture. Let us further generalize this probability path and associated rate to be non-factorized, applying auxiliary variables again. We first re-express this probability path through an auxiliary boolean variable $\mathsf{m} \in \{\texttt{false}, \texttt{true}\}^N$:

$$p_t(z|z_0, z_1) = \sum_{\mathsf{m}} p_t(\mathsf{m}|z_0, z_1)p_t(z|\mathsf{m}, z_0, z_1), \tag{37}$$

$$\text{where } p_t(z|\mathsf{m}, z_0, z_1) = \prod_{i=1}^N \mathbb{1}_{[\neg \mathsf{m}^i]}\delta_{z_0^i}(z^i) + \mathbb{1}_{[\mathsf{m}^i]}\delta_{z_1^i}(z^i), \tag{38}$$

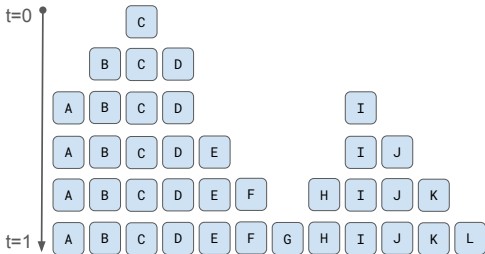

Figure 8: Illustration of the localized generation path. Tokens that are neighboring to existing tokens have a much higher likelihood of getting inserted next. Tokens that are not next to already inserted tokens (e.g. token **I**) have a small, but non-zero likelihood of getting inserted in the current timestep.

where $\mathbb{1}_{[\cdot]}$ is the indicator function and returns one if the input is $\texttt{true}$, zero otherwise. That is, $\mathsf{m}^i$ indicates whether $z^i$ is equal to $z_0^i$ or $z_1^i$. In the case of $p_t(\mathsf{m}|z_0, z_1)$ being be a factorized distribution, this would recover the factorized probability path (8).

$$p_t(\mathsf{m}|z_0, z_1) = \prod_{i=1}^{N} p_t(\mathsf{m}^i|z_0, z_1), \qquad p_t(\mathsf{m}^i|z_0, z_1) = \mathbb{1}_{[\neg\mathsf{m}^i]}(1 - \kappa_t) + \mathbb{1}_{[\mathsf{m}^i]}\kappa_t \qquad (39)$$

This helps ensure that the conditional rates can be constructed easily. However, this could be problematic for Edit Flows as when the sequence length becomes large, noisy sequences $x_t$ will consist of non-neighboring tokens. Instead, we will propose a non-factorized locality-based construction in which if $\mathsf{m}^j$ is $\texttt{true}$, it incites nearby values ($\mathsf{m}^{j-1}$ and $\mathsf{m}^{j+1}$) to transition their value to $\texttt{true}$, thereby encouraging nearby neighbors to have similar values.

Let us consider an extended space of boolean variables denoted by $\mathsf{M} \in \{\texttt{true}, \texttt{false}\}^{N \times N}$ and consider $N$ independent CTMC processes, starting at all values being $\texttt{false}$. For each row $\mathsf{M}^i$, we create a process where $\mathsf{M}^{i,i}$ first switches to $\texttt{true}$ according to a time-dependent rate $\lambda_t^{\mathrm{indep}}$ and this then propagates to neighboring values according to a propagation rate $\lambda^{\mathrm{prop}}$. This can be concisely expressed as the following CTMC process for each $\mathsf{M}^i$.

$$u_t(\mathsf{M}^{i,j}|\mathsf{M}_t^i) = \left(\lambda_t^{\mathrm{indep}}\delta_{ij} + \mathbb{1}_{[\mathsf{M}_t^{i,j-1} \vee \mathsf{M}_t^{i,j+1}]}\lambda^{\mathrm{prop}}\right)\left(\mathbb{1}_{[\mathsf{M}^{i,j}]} - \delta_{\mathsf{M}_t^{i,j}}(\mathsf{M}^{i,j})\right), \qquad \mathsf{M}_0^i = \texttt{false}, \tag{40}$$

where $\delta_{ij} = 1$ if $i = j$ and $\delta_{ij} = 0$ otherwise. Breaking this down, $\lambda_t^{\mathrm{indep}}$ is an independent rate for switching $\mathsf{M}^{i,i}$ to $\texttt{true}$ regardless of the value of $\mathsf{M}_t$ at other positions—if we only have this independent part, then this formulation will be equivalent to the factorized case—and $\lambda^{\mathrm{prop}}$ is the rate for the off-diagonals $\mathsf{M}^{i,j}$ if a neighbor is $\texttt{true}$, responsible for propagating along local neighborhoods—for simplicity, this part is time independent. We then map from this extended space to the space of $\mathsf{m}$ by the mapping:

$$\mathsf{m}_t^j = \mathsf{M}_t^{1,j} \vee \mathsf{M}_t^{2,j} \vee \cdots \vee \mathsf{M}_t^{N,j} \tag{41}$$

That is, $\mathsf{m}_t^j$ is $\texttt{true}$ if any value in the column of $\mathsf{M}_t^{:,j}$ is $\texttt{true}$.

**Augmented rate.** We now have a rate $u_t(\mathsf{M}|\mathsf{M}_t, z_0, z_1)$ that generates $p_t(\mathsf{M}|z_0, z_1)$ and can apply Lemma B.2 twice to determine rate $u_t(z, \mathsf{m}, \mathsf{M}|z_t, \mathsf{m}_t, \mathsf{M}_t, z_0, z_1)$ that generates $p_t(z, \mathsf{m}, \mathsf{M}|z_0, z_1)$. The target summed rate we need for training a localized path model (where we consider $z$ as observed and $(\mathsf{m}, \mathsf{M})$ as auxiliary) is for $z \neq z_t$

$$\sum_{\mathsf{m}, \mathsf{M}} u_t(z, \mathsf{m}, \mathsf{M}|z_t, \mathsf{m}_t, \mathsf{M}_t, z_0, z_1) \tag{42}$$

$$= \begin{cases} \delta_{z_1^j}(z^j)\left(\lambda_t^{\mathrm{indep}} + \sum_i \mathbb{1}_{[\mathsf{M}_t^{i,j-1} \vee \mathsf{M}_t^{i,j+1}]}\lambda^{\mathrm{prop}}\right) & \text{if } z \text{ and } z_t \text{ differ only in } j\text{-th token} \\ 0 & \text{otherwise} \end{cases} \tag{43}$$

To utilize specifically for localized edit flows, we extend our rates again to generate $p_t(x, z, \mathsf{m}, \mathsf{M}|z_0, z_1)$ and the rate needed for training localized edit flows, prior to the sum over

additional auxiliary $z$, is simply (42) multiplied by $\delta_{f_{\text{rm-blanks}}(z)}(x)$. Following the same steps as before, the edit flow training loss using localized rates is:

$$\mathcal{L}(\theta, \lambda) = -\mathbb{E}_{t, \pi(z_0, z_1), p_t(x_t, z_t, \mathsf{m}_t, \mathsf{M}_t | z_0, z_1)} \left[ u_t^\theta(x_t | x_t) + \sum_{i=1}^{N} \mathbb{1}_{[z_1^i \neq z_t^i]} \lambda_{i,t}^{\text{eff}} \log u_t^\theta(x(z_t, z_1, i), x_t) \right]$$
(44)

where $x(z_t, z_1, i) = f_{\text{rm-blanks}}(z_t^1, \ldots, z_t^{i-1}, z_1^i, z_t^{i+1}, \ldots, z_t^N)$ and $\lambda_{i,t}^{\text{eff}} = \lambda_t^{\text{indep}} + \sum_l \mathbb{1}_{[\mathsf{M}_t^{l,i-1} \vee \mathsf{M}_t^{l,i+1}]} \lambda^{\text{prop}}$.

**Parameterization.** For $\lambda_t^{\text{indep}}$, we can reuse the same form from the factorized case defined by a scheduler $\kappa_t$,

$$\lambda_t^{\text{indep}} = \frac{\dot{\kappa}_t}{1 - \kappa_t}, \quad \kappa_0 = 0, \kappa_1 = 1$$
(45)

which allows us to ensure that $\mathsf{m}_1^i = \texttt{true}$ for all $i$ and whose integral can be obtained easily. For $\lambda^{\text{prop}}$, we choose an appropriate constant, the value of which corresponds to the expected number of propagations within a unit interval of time.

**Sampling.** In order to allow efficient training, we need to sample $(\mathsf{m}_t, \mathsf{M}_t)$ for a given $t$ without simulating the CTMC (40). The construction of (40) is designed explicitly to allow efficient sampling. Since the CTMC processes are independent for each $\mathsf{M}^i$, we can simulate them independently. Furthermore, for every $\mathsf{M}^{i,j}$ the source of the propagation can only be from $\mathsf{M}^{i,i}$. Thus, we can make use of the following 2-step sampling algorithm given $t$:

1. For each $i$, independently sample the time $t_i^* \in [0, 1]$ that each $\mathsf{M}^{i,i}$ would switch to $\texttt{true}$ based on the independent rate $\lambda_t^{\text{indep}}$. If $t_i^* \leq t$, then $\mathsf{M}_t^{i,i}$ is set to $\texttt{true}$.

2. For each $i$ such that $t_i^* \leq t$, sample the number of neighbors to the left and right that are switched to $\texttt{true}$ due to propagation with rate $\lambda^{\text{prop}}$ from $\mathsf{M}^{i,i}$ during the time interval $[t_i^*, t]$.

Afterwards, we can set $\mathsf{m}_t^j = \mathsf{M}_t^{1,j} \vee \mathsf{M}_t^{2,j} \vee \cdots \vee \mathsf{M}_t^{N,j}$.

*Step 1* of this sampling algorithm requires determining the time of the switch $t^*$. This is equivalent to finding the occurrence time of an inhomogeneous Poisson process with intensity function $\lambda_t^{\text{indep}}$. This can be done via the inverse method (Rasmussen, 2018) as follows.

1. Sample $u \sim \text{Unif}(0, 1)$

2. Compute $t^*$ s.t. $u = \exp\{-\int_0^{t^*} \lambda_t^{\text{indep}} dt\}$

For the parameterization in (45), we can analytically derive this.

$$t^* = \kappa^{-1}(u) \quad \text{where} \quad u \sim \text{Unif}(0, 1)$$
(46)

*Step 2* of the sampling algorithm consists of determining how many neighbors get propagated from each source $\mathsf{M}^{i,i}$ within a certain time interval $[t_i^*, t]$. Since neighbors on the same side can only get propagated sequentially, this is equivalent to determining the number of occurrences from a homogeneous Poisson process with intensity $\lambda^{\text{prop}}$. The formula for this is

$$\mathsf{N}_i \sim \text{Pois}(\cdot \; ; \; \lambda^{\text{prop}} \Delta t_i), \quad \text{where} \quad \Delta t_i = t - t_i^*$$
(47)

We would sample two variables i.i.d. $\mathsf{N}_i^l$ and $\mathsf{N}_i^r$ for the number of neighbors propagated to the left and to the right of $\mathsf{M}^{i,i}$, respectively. The logic for $\mathsf{M}_t^{i,j}$ can be concisely expressed as

$$\mathsf{M}_t^{i,j} = \mathsf{M}_t^{i,i} \wedge \left[ \left( (j < i) \wedge (j \geq i - \mathsf{N}_i^l) \right) \vee \left( (j > i) \wedge (j \leq i + \mathsf{N}_i^r) \right) \right].$$
(48)

All computations within each step of the sampling algorithm can be completely parallelized, resulting in fast sampling of $\mathsf{m}_t$.

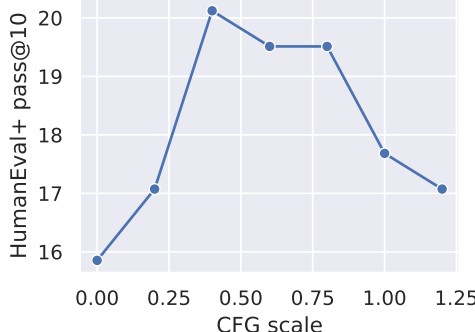
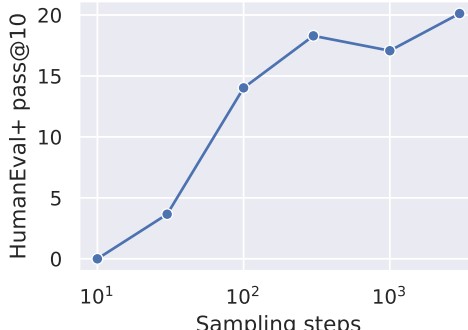

Figure 11: Effect of CFG scale at sampling time on the code generation benchmark using the 1.3B parameter Edit Flow model.

Figure 12: Effect of the number of sampling steps on the code generation benchmark using the 1.3B parameter Edit Flow model.

## D Training data analysis

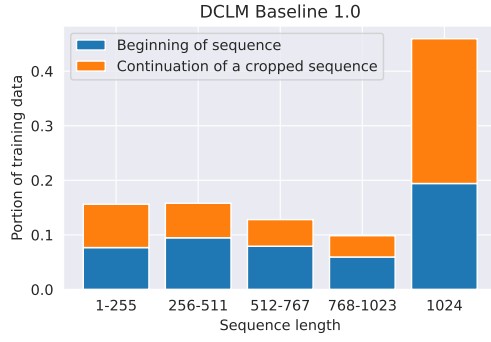
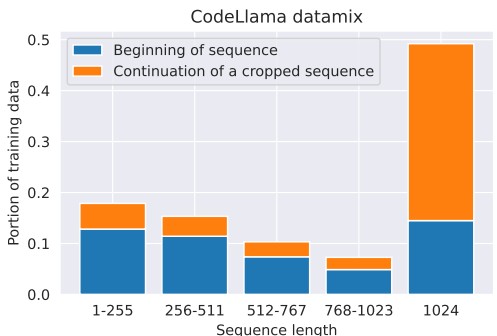

Figure 9: 54% of the training data consists of sequences of length < 1024 and 57% of these are self contained sequences (meaning that they start with a <BOS> token and have < 1024 tokens in total).

Figure 10: 50% of the training data consists of sequences of length < 1024 and 72% of these are self contained sequences (meaning that they start with a <BOS> token and have < 1024 tokens in total).

## E Further experimental details

**Training:** All models were trained of 500,000 steps with batch size of 4096 distributed across $16 \times 8$ H100 GPUs, which resulted in ~2T tokens used for the Autoregressive and Mask DFM models. Since the Edit models do not use compute for tokens that are missing from the sequence, they are considerably more compute efficient. They were able to ingest ~6T tokens during the same 500,000 training steps.

**Architecture:** Table 4 shows the details of the architecture and optimizer used in our experiments.

**Conditioning:** A beginning of each sequence in the training set is designated to be conditioning. The portion of the sequence used as conditioning is randomly chosen to be $c^3$ where $c \sim U[0,1]$. For 10% of the sequences, we drop the conditioning to allow for unconditional prediction and CFG scaling at inference time.

**Image conditioning:** To condition our model on an image input, we follow Liu et al. (2024) and use an early fusion approach of appending image embeddings as prompts to our sequence models.

| Hyperparameter | 280M configuration | 1.3B configuration |
|---|---|---|
| Vocabulary size | 32k | 32k |
| Model dimension | 1024 | 2048 |
| Conditioning dimension | 32 | 64 |
| Number of layers | 12 | 16 |
| Number of heads | 16 | 32 |
| Number of KV heads | 8 | 8 |
| Feed-forward dimension | 1740 | 3072 |
| Feed-forward hidden dimension | 6963 | 12288 |
| Training steps | 500k | 500k |
| Batch size | 4096 | 4096 |
| Optimizer | AdamW | AdamW |
| Learning rate | 3e-4 | 3e-4 |
| Beta 1 | 0.9 | 0.9 |
| Beta 2 | 0.95 | 0.95 |
| Warmup steps | 2000 | 2000 |
| Learning rate schedule | cosine | cosine |

Table 4: Details of the Llama3 architecture and optimizer used in our experiments. Conditioning dimension is used in the text and code experiments: it denotes the dimensionality of an the embedding carrying the binary signal whether a given token is part of the conditioning or not.

| Sampler Hyperparameter | Autoregressive | | Mask DFM | | Edit Flow | | Uniform $X_0$ + Edit Flow | |
|---|---|---|---|---|---|---|---|---|
| | Pass@1 | Pass@10 | Pass@1 | Pass@10 | Pass@1 | Pass@10 | Pass@1 | Pass@10 |
| Sampling steps | | | 1000 | 1000 | 10000 | 5000 | 5000 | 5000 |
| Classifier-free guidance | | | 1.5 | 1.5 | 0.5 | 0.5 | 0.5 | 1.0 |
| Temperature | 0.0 | 1.0 | 0.8 | 0.8 | 0.8 | $0.8t+1.0(1-t)$ | $0.8t+1.0(1-t)$ | 0.8 |
| Divergence-free component | | | $5t^{0.25}(1-t)^{0.5}$ | $10t^{0.25}(1-t)^{0.5}$ | $60t^{1.5}(1-t)^{0.5}$ | $150t^{1.0}(1-t)^{0.25}$ | $10.0t^{0.25}(1-t)^{0.25}$ | $10.0t^{0.5}(1-t)^{1.0}$ |
| Top p | 0.0 | 0.7 | - | - | 0.5 | 0.3 | 0.7 | 0.9 |
| Top k | 1 | - | 2 | 2 | - | - | - | - |
| Reverse CFG | | | | | | | -0.5 | -1.0 |
| Reverse temperature | | | | | | | 0.5 | 0.2 |
| Reverse top p | | | | | | | 0.8 | 0.8 |

Table 5: Sampling parameters used in the code experiments. The parameters were tuned by running random search (N=200 runs for pass@1 and N=20 runs for pass@10) on the HumanEval benchmark. The HumanEval results were then re-computed using a new random seed to avoid evaluation set leakage.

We use frozen CLIP embeddings (Radford et al., 2021) for computing image embeddings and then map it to the same dimension as the sequence model with a 1-layer MLP projector.

**Sampling:** For the pass@1 and pass@10 benchmarks, we tuned the sampling parameters (temperature, top_p, sampling steps, CFG, divergence-free component) for each model separately with the goal of maximizing performance. Figures 11 and 12 show the impact of CFG scale and the number of sampling steps on generation quality. Table 5 shows the sampling parameters used for evaluation in the code benchmarks.

**Mask DFM:** The Mask DFM baseline is trained using the ELBO objective (Shaul et al., 2024) in the image captioning experiments and using the cross-entropy objective (Gat et al., 2024) in the code and text experiments. Training data that does not meet the sequence length 1024 used by the model is padded using a padding token. This padding token, if generated by the model, is removed at inference time.

**Text benchmarks:** Table 6 shows the CFG scales tuned for the text benchmarks.

| Method | HellaSwag | ARC-E | ARC-C | PIQA | OBQA | WinoGrande |
|---|---|---|---|---|---|---|
| Mask DFM | 0.0 | 0.5 | 0.0 | 0.5 | 0.0 | 0.0 |
| Edit Flow | 1.0 | 0.5 | 0.5 | 0.5 | 1.0 | 0.5 |

Table 6: CFG scales used in the text benchmarks. We only tuned CFG scale: we swept the values 0.0, 0.5, 1.0, 2.0, 5.0 and 10.0 on every benchmark and report the best results.

## F  Model preference for minimal edits

Similar to continuous flow matching, the generated coupling $p_1(x_1|x_0)$ may differ from the coupling used during training, denoted as $\pi(x_1|x_0)$. The model learns a coupling that involves fewer edits than the average observed training. To illustrate this, we applied edit flows to a toy dataset that includes only insert and delete operations, with no substitutions. The distributions of $\pi(x_0)$ and $\pi(x_1)$ are both uniform over strings of length 4 containing only the characters A and B (as shown in Figure 14). The probability path is defined such that every character in $x_0$ gets deleted and every character in $x_1$ gets inserted (least optimal alignment). The coupling at training time is uniform.

However, the model does not retain the uniform coupling from training. Figure 14 demonstrates that it prioritizes $x_0, x_1$ pairings that require the fewest edits. For example, $x_0 = AAAA$ is $20\times$ more likely to generate $x_1 = AAAA$ (requiring no edits) than $x_1 = BBBB$ (requiring 4 insertions and 4 deletions). Generally, the cells with the highest values of $p_1(x_1|x_0)$ correspond to pairings that require only a few edits, while the lowest values correspond to pairings that require many edits.

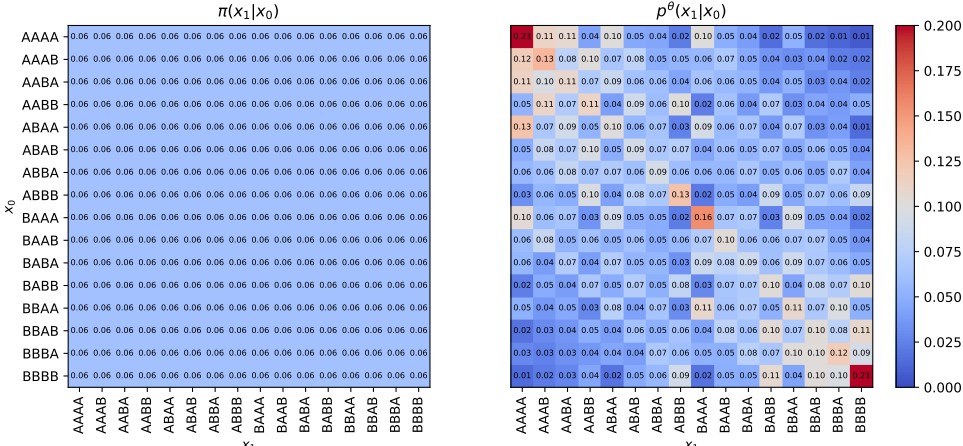

Figure 14: Comparison of the training time coupling ($\pi(x_1|x_0)$) with the coupling learned by the edit flow ($p^\theta(x_1|x_0)$). The model prioritizes pairings that require few edits.

```python
def get_z(ids: list[int]) -> tuple[list[int], list[int]]:
    num_substitutions = min(len(ids), target_num_substitutions)
    num_deletions = target_num_deletions + target_num_substitutions - num_substitutions
    x_0 = x_0 = np.random.randint(low=0, high=vocab_size, size=num_deletions + num_substitutions).tolist()
    sub_id = 0
    z = (
        [epsilon_0_id] * (len(ids) - num_substitutions)
        + [epsilon_1_id] * num_deletions
        + [sub_id] * num_substitutions
    )
    random.shuffle(z)

    z_0: list[int] = []
    z_1: list[int] = []
    ids_index = 0
    x_0_index = 0
    for token in z:
        if token == epsilon_1_id:
            z_0.append(x_0[x_0_index])
            z_1.append(epsilon_1_id)
            x_0_index += 1
        elif token == epsilon_0_id:
            z_0.append(epsilon_0_id)
            z_1.append(ids[ids_index])
            ids_index += 1
        elif token == sub_id:
            z_0.append(x_0[x_0_index])
            z_1.append(ids[ids_index])
            x_0_index += 1
            ids_index += 1
    return z_0, z_1

def get_z_t(z_0: list[int], z_1: list[int], kappa: float) -> list[int]:
    return [
        token_0 if np.random.uniform() > kappa else token_1
        for token_0, token_1 in zip(z_0, z_1)
    ]

# Training loop
for sample in training_samples:
    tokens: list[int] = encode(sample, bos=False)
    z_0, z_1 = get_z(tokens)
    z_0 = [bos_id] + z_0
    z_1 = [bos_id] + z_1
    t: float = np.random.uniform()
    kappa: float = t  # Using a linear schedule
    dkappa: float = 1.0
    z_t: list[int] = get_z_t(z_0, z_1, kappa)
    x_t: list[int] = remove_epsilon(z_t)
    x_t_tensor: torch.Tensor = torch.tensor(x_t).to(device)

    # Forward pass
    insert_lambda, insert_q, delete_lambda, substitute_lambda, substitute_q = model(
        x_t_tensor, t
    )

    # Calculate loss
    loss_term_1: torch.Tensor = torch.sum(
        insert_lambda + delete_lambda + substitute_lambda
    )
    loss_term_2: torch.Tensor = torch.tensor(0.0, device=device)
    x_t_index: int = -1  # Corresponding index in x_t
    for token_t, token_1 in zip(z_t, z_1):
        if token_t != epsilon_0_id and token_t != epsilon_1_id:
            x_t_index += 1

        if token_t == epsilon_0_id and token_1 != epsilon_1_id:
            # Missing token must be inserted
            loss_term_2 = loss_term_2 - (dkappa / (1 - kappa)) * torch.log(
                insert_lambda[x_t_index] * insert_q[x_t_index, token_1]
            )
        elif token_t != epsilon_0_id and token_1 == epsilon_1_id:
            # Extra token must be deleted
            loss_term_2 = loss_term_2 - (dkappa / (1 - kappa)) * torch.log(
                delete_lambda[x_t_index]
            )
        elif (
            token_t != epsilon_0_id
            and token_1 != epsilon_1_id
            and token_t != token_1
        ):
            # Incorrect token must be substituted
            loss_term_2 = loss_term_2 - (dkappa / (1 - kappa)) * torch.log(
                substitute_lambda[x_t_index] * substitute_q[x_t_index, token_1]
            )

    loss: torch.Tensor = loss_term_1 + loss_term_2
    optimizer.zero_grad()
    loss.backward()
    optimizer.step()
```

Figure 13: Simplified training code for Edit Flows. The helper functions `get_z` and `get_z_t` generate noisy and target token sequences, while the training loop computes the loss and updates the model parameters. For brevity, we did not include features such as batching, conditioning on a random portion of the sequence and scaling the model outputs by the rate.

