# OpenReview forum: "Edit Flows: Variable Length Discrete Flow Matching with Sequence-Level Edit Operations"
_NeurIPS.cc/2025/Conference — NeurIPS 2025 poster_

### Official Review · Reviewer_czyg · 2025-06-21

**Clarity:** 2
**Significance:** 2
**Originality:** 2
**Rating:** 3
**Confidence:** 3

**Summary:**

Problem: Non-autoregressive models struggle with variable-length sequence generation, unlike autoregressive models.
Proposed solution: Introduce Edit Flows, a non-autoregressive model using edit operations (insertions, deletions, substitutions). This allows position-relative, non-rigid generation, better matching real sequence structure. For training this, they use an alignment between start and end states to form data that they then train on. They also use auxiliary variables (epsilon) to denote empty words after deletion or before insertion.

**Questions:**

I think The real details actually start from line 170. Up to this point, it's all formalisms about continuous processes and how they may connect to discrete time.


Training
=======
Here is my understanding (and let me know if I am understanding it correctly). To train this model to map x_0 to x_1, you want to create a path that connects the starting point to the finish line. And for doing this, you do some manual alignment. For example, as your example shows, you use the path characterized by edit distance between two sequences.

If the above is correct, you should probably discuss this work that has a similar data generation pipeline and contrast against it: https://arxiv.org/abs/2211.00053

One thing that I don't fully understand: do you think this edit-distance-based data augmentation can be done for all tasks?


Experiments ;
=========
I am not convinced your baselines are actually strong. For example, for OBQA and ARC-C (Table 2), the numbers are barely above random (25%). Meanwhile, we know that the scores on these datasets have been much higher as far as we know, even with. 1B model.

One other issue: why do these tasks require sequence editing? To be honest, I don't have any concrete suggestions. But I do wonder if there are more appropriate choices.



Other works
===========
My suggestion is to move the related work to the end of your paper to not create a break between formalism and experiments.

Other missing citations:
https://arxiv.org/pdf/1811.10996
https://arxiv.org/abs/2202.11705
https://aclanthology.org/2020.emnlp-main.701.pdf

**Ethical Concerns:**

["NO or VERY MINOR ethics concerns only"]

**Final Justification:**

First, I want to acknowledge the theoretical contribution of this paper (as I have in my review).
I remain unconvinced by the empirical results.

**Limitations:**

Yes.

**Paper Formatting Concerns:**

Nope

**Quality:**

3

**Strengths And Weaknesses:**

- I am convinced by the motivation: we need to go beyond auto-regressive (one-directional) generation/sampling.
- However, I am not convinced whether the proposed framework/algorithm is adding anything to our existing knowledge. If you distill the idea, it all comes down to curating training data for text editing and training on it; it's a pretty simple framework that is wrapped inside an esoteric formalism.
- I am also not convinced whether the proposed framework is adding anything to our existing empirical toolkit. The experimental setup is weak, to my understanding.

---

> ### Author Rebuttal · Authors · 2025-07-30
>
> We thank the reviewer for their review and for going through the paper despite difficulty in understanding our mathematical notation, which we found very unfortunate but acknowledge the feedback. However, we believe that what we proposed are significant modeling improvements beyond simple data manipulation, and we hope that the following statements may help clarify our contributions:
>
>   - Unlike the works that the reviewer brought up [1,3,4], we do not provide extra knowledge to the model (such a scoring function for intermediate sentences, or an ordered sequence of target edit operations). Our models only see the data sequences, and either (i) deletes all tokens or (ii) performs completely random edit operations to transform to a noise distribution. In either case, we simply map from a noise distribution (the null sequence or a sequence of random tokens) to the data sequence, and there is no external knowledge given to the model.
>
>   - The alignments that the reviewer refers to as data processing (line 170) is actually only for training tractability. We use this to align the model's outputs (which has a shorter length due to deletion of tokens) back to the original sequence (which contains all tokens). In practice, these alignments are sampled randomly (e.g., from randomly deleting tokens independently) and do not provide the model with additional information.
>
>   - The insert-only Edit Flow model sees the *exact* same data and noise sequences as the mask diffusion baseline (denoted Mask DFM). The core difference is only whether we actually delete a token (Edit Flow) or replace it with a `<MASK>` token (Mask DFM). As such, any improvements we see are due to the differences in modeling paradigms, not data processing.
>
>   - The complexity in the  formalism is in order to model the stopping criterion "when to stop editing". Unlike the works that the reviewer brought up [2,3], we do not use a non-terminating MCMC algorithm but model a terminating process. As such, the model automatically *learns when edits should be done* and stops when no more edits are needed. These are the rates denoted as $\lambda$, which have not been used by most prior works. Prior methods used different stopping criterion because they were restricted models (e.g., AR uses EOS token but are limited to left-to-right generation, mask diffusion stops when all tokens are unmasked but are limited to fixed sequence lengths).
>
> We hope the above statements help clarify our contributions. We understand that the continuous-time Markov chain formalism is new to the community and had hoped to introduce and extend it as gently as possible.
>
> Below, we answer the specific questions.
>
> > I am not convinced whether the proposed framework/algorithm is adding anything to our existing knowledge. If you distill the idea, it all comes down to curating training data for text editing and training on it; it's a pretty simple framework that is wrapped inside an esoteric formalism.
>
> Firstly, note that our proposed model has the ability to generate variable length sequences in a non-autoregressive fashion, which is not possible with the Mask DFM construction.
>
> Secondly, we note that introducing edit operations in the generation process (see line 140) introduces interesting behaviors. The Edit Flow model predicts a set of tokens at to be inserted at each location; however, the model doesn't need to be optimal and can predict any one of the tokens within this set since the generation process can insert the other tokens at a later time. This allows the model to generate whatever token it is most confident about and then self-correct (e.g. fix the grammar) later on. We are including additional qualitative samples in the appendix to showcase this generative behavior.
>
> > I am also not convinced whether the proposed framework is adding anything to our existing empirical toolkit.
>
> Our experiments are aimed at an apples-to-apples comparison between modeling frameworks. These ablation studies are vital in understanding the tradeoffs between different frameworks. Even at the 1B scale, the text and code models already take weeks to train, making understanding their design choices crucial prior to scaling up.
>
> > [paraphrased] Confirmation of understanding: The training involves creating a path connecting start to finish using manual alignment, e.g., edit distance between sequences. Is this correct?
>
> We first note that we are mapping from a completely noise distribution (either null sequence or completely random tokens) to a data sequence. The alignments are completely random and are not user-provided manually-tuned edit sequences. For instance, they do not correspond to the optimal alignment using edit distance as it wouldn't make sense with one sequence having random tokens. We believe this may be a source of misunderstanding, as some prior works have trained on curated sequences of edits; *our paper is not about curating sequences of edit operations*.
>
> > [...] path characterized by edit distance between two sequences
>
> To satisfy our curiosity, we tested the reviewer's suggestion of using edit distance to produce an optimal alignment between a model-generated sequence and the data sequence (this new experiment does use additional knowledge from a pretrained model but does not impose any priorities on the edits). Surprisingly, we found that this did not improve but hurt performance. We believe it is because the model does not see diverse enough sequences during training.
>
> > If the above is correct, you should probably discuss this work that has a similar data generation pipeline and contrast against it: [1]
>
> Thank you for bringing our attention to this work and the other works you mentioned; we will update the related work section. In the referenced work, one model proposes complete sentences and another model scores the sentences. In contrast, Edit Flows do not require a separate base-generator and a corrector model; a single model is used to generate the samples. Our generated samples are also not necessarily complete or coherent sentences until the final generation step. We added more qualitative examples in the appendix to illustrate this.
>
> > One thing that I don't fully understand: do you think this edit-distance-based data augmentation can be done for all tasks?
>
> Since we do not introduce additional knowledge, the framework works for any generative sequence modeling task.
>
> > I am not convinced your baselines are actually strong. For example, for OBQA and ARC-C (Table 2), the numbers are barely above random (25%). Meanwhile, we know that the scores on these datasets have been much higher as far as we know, even with. 1B model.
>
> Thank you for bringing this to our attention. If the reviewer is aware of a 1B model trained on the DCLM-baselines-1.0 dataset only, please let us know so we can cross reference. All of our reported results are zero-shot, whereas many prior works used few-shot evaluation or performed model finetuning. The goal of our experiments is a systematic study of the different modeling paradigms, aimed at improving our understanding.
>
> Thanks to the reviewer's comments, we investigated our DCLM results and we found that our models slightly underperform on these benchmarks because we were training them only on a subset of DCLM. We are correcting this and we will update the results. We emphasize that the experiment still serves its objective as an ablation study. All models were trained on the same data using the same resources, therefore we do not expect the conclusions to change.
>
> > One other issue: why do these tasks require sequence editing? To be honest, I don't have any concrete suggestions. But I do wonder if there are more appropriate choices.
>
> We incorporate edit operations mainly to handle variable length generation with non-autoregressive models, which are required for most text generation tasks such as captioning and code generation.
>
> > Other missing citations: [...]
>
> Thank you for bringing these works to our attention. We will include them in our related works section; a summary of their differences:
> [2] is similar to Edit Flows in the sense that the model makes predictions in the space of edits. A key difference, however, is that Edit Flows sample from the target distribution in a fixed number of steps in the Flow Matching framework, as opposed to [2] that generates samples from a stationary distribution until acceptance.
> [3] and Edit Flows both iteratively improve on the sentence until the task is complete, however Edit Flows make predictions in the space of token edits, whereas [3] predicts the next sequence in the generation process as a whole.
> [4] makes predictions in the space of edits, however, unlike Edit Flows, [4] requires a differentiable objective function that scores language fluency and semantic meaning. Edit Flows only require a single model and does not use backpropagation at test time.
>
> Please let us know if you have additional questions. We are happy to further clarify our contributions, and hope to rectify any misunderstanding.
>
> [1] Welleck, Sean, et al. "Generating Sequences by Learning to Self-Correct." The Eleventh International Conference on Learning Representations.
>
> [2] Miao, Ning, et al. "Cgmh: Constrained sentence generation by metropolis-hastings sampling." Proceedings of the AAAI Conference on Artificial Intelligence. Vol. 33. No. 01. 2019.
>
> [3] Qin, Lianhui, et al. "Cold decoding: Energy-based constrained text generation with langevin dynamics." Advances in Neural Information Processing Systems 35 (2022): 9538-9551.
>
> [4] Sha, Lei. "Gradient-guided unsupervised lexically constrained text generation." Proceedings of the 2020 Conference on Empirical Methods in Natural Language Processing (EMNLP). 2020.
>
> List of manuscript changes in response to the feedback:
> 1. Expanded the related works section to include the four missing references.
> 2. Added qualitative examples to showcase the Edit Flow generation process.

---

> > ### Comment · Reviewer_czyg · 2025-08-05
> >
> > #
> >
> > Apologies for slow response. I needed to set aside some time to carefully go through your comments, draft and other comments.
> >
> > Here are a summary of my remaining issues (you can see the details in the later text):
> >
> > - The empirical gains:
> >     - Are the significant?
> >     - Are they consistent across different models?
> >     - Do they generalize to longer sequences?
> > - The conceptual understand: what you adding to the literature?
> >
> > ---
> >
> > Responses to my comments:
> >
> > > Unlike the works that the reviewer brought up [1,3,4], we do not provide extra knowledge to the model
> > >
> >
> > The most of the prior work also **do not necessitate** external/additional knowledge. The incorporation of additional knowledge is an added feature.
> >
> > This also leads me to your response on empirical results:
> >
> > >
> > >
> > >
> > > > I am not convinced your baselines are actually strong. For example, for OBQA and ARC-C (Table 2), the numbers are barely above random (25%). Meanwhile, we know that the scores on these datasets have been much higher as far as we know, even with. 1B model.
> > > >
> > >
> > > Thank you for bringing this to our attention. If the reviewer is aware of a 1B model trained on the DCLM-baselines-1.0 dataset only, please let us know so we can cross reference. All of our reported results are zero-shot, whereas many prior works used few-shot evaluation or performed model finetuning. The goal of our experiments is a systematic study of the different modeling paradigms, aimed at improving our understanding.
> > >
> >
> > Then how does the model know what task you want it to do? Ultimately, you need some sort of lever to steer the systems. What is that “lever”? This makes me wonder if the reported results are purely doe to random chance. For your results in table 2,3, can you please provide their confidence intervals with Bootstrap resampling?
> >
> > I see why you’re avoiding “few-shot” setup: you’re starting from noise/null sequence and hence, it does seem (to my understanding) But by doing so, you do not give your model any ability be controlled. While you may consider “few-shot” eval unfair, others may consider it a fair game: first, coming up with 3-4 examples it not a big deal (I can do it in 10 mins). Plus, auto-regressive models have out-of-box in-context learning (you don’t need to train them for it).
> >
> > I am going to re-iterate this: “OBQA and ARC-C (Table 2), the numbers are barely above random (25%).” Why should I believe that your numbers/gains any meaningful?
> >
> > > The alignments are completely random and are not user-provided manually-tuned edit sequences. For instance, they do not correspond to the optimal alignment using edit distance as it wouldn't make sense with one sequence having random tokens. We believe this may be a source of misunderstanding, as some prior works have trained on curated sequences of edits; *our paper is not about curating sequences of edit operations*.
> > >
> >
> > Yes, thank you for clarifying this. Appreciate it.
> >
> > > In practice, these alignments are sampled randomly (e.g., from randomly deleting tokens independently) and do not provide the model with additional information.
> > >
> >
> > This is confusing to me a bit. When I look at the sampled generations in Figure 8, there is a sense in which the model starts with generating most important/informative tokens (e.g, “black” for the picture of the dog with a black hat on). Alternatively, perhaps the examples are cherry-picked?
> >
> > > The complexity in the formalism is in order to model the stopping criterion "when to stop editing". Unlike the works that the reviewer brought up [2,3], we do not use a non-terminating MCMC algorithm but model a terminating process. As such, the model automatically *learns when edits should be done* and stops when no more edits are needed. These are the rates denoted as , which have not been used by most prior works. Prior methods used different stopping criterion because they were restricted models (e.g., AR uses EOS token but are limited to left-to-right generation, mask diffusion stops when all tokens are unmasked but are limited to fixed sequence lengths).
> > >
> >
> > Can you point me to where this is discussed in your text?
> >
> > > In contrast, Edit Flows do not require a separate base-generator and a corrector model; a single model is used to generate the samples.
> > >
> >
> > Yes I understand this. But the separation of roles in this reference is a shallow one and not a crucial aspect of the work, to my limited understanding. The main thesis here is casting generation as an iterative refinement, akin to yours.
> >
> > I do think that the references [2,4] are also quite similar to some of your key contributions in terms of incremental text editing/addition/deletion. What do you think?
> >
> > Other questions:
> >
> > - Can you please share a few examples of your training data?
> > - Can you please share a few lines of your code pertain to your training objective? (I see the equations but I’d like to see how you’ve implemented it)

---

> > > ### Comment · Reviewer_czyg · 2025-08-05
> > >
> > > On your exchange with `arpi` and `LxY230`:
> > >
> > > >
> > > >
> > > >
> > > > > It would be nice to see some larger and more recent image captioning models, to compare best-on-best performance (though this may need to be independent of model size). As it stands, the qualitative examples in the app
> > > > >
> > > >
> > > > We note that the primary value of all of our experiments is to analyze the ability of the models on **their own when trained from scratch**, without additional data or pretrained weights. This allows us to examine model performance differences without confounding variables such as varying training data. Most papers that report image captioning use pretrained weights (e.g. VLP and ClipCap which we included for reference).
> > > >
> > >
> > > The reviewer is also saying this too: “their own when trained from scratch”. They’re not [necessarily] suggesting that you give additional information to the model. But rather, the reviewer’s concern is whether your gains are limited to weak models.
> > >
> > > Later you say that:
> > >
> > > > However, to satisfy our and the reviewer's curiosity, we have conducted an additional experiment where we initialize the weights using a Llama 3 1B model, as a means to test how well Edit Flows can adopt the weights of a pretrained AR model trained on internet-scale data. This increased our CIDEr score on MSCOCO from 108 to 124, which now outperforms both VLP and ClipCap (a 1.5B **GPT-2** model).
> > > >
> > >
> > > Well, this is clearly an unfair comparison to contrast llama3 vs GPT-2. You’d need to repeat the **whole table for** llama3 for apples to apples comparison.   The reviewer is asking you to repeat the same table, for a fixed model (for difference choices of models).
> > >
> > > > However, we emphasize that the value of our experiments is in having a concrete apples-to-apples comparison on the same data and compute scale.
> > > >
> > >
> > > Yes, and the reviewer does not disagree. They’re asking for this apples-to-apples comparison for difference choices of compute budget.
> > >
> > > ---
> > >
> > > On the exchange with `C6NG02`
> > >
> > > > we did not attempt evaluating the model on longer than 1024 sequence length, though we imagine it would not perform well compared to shortening the prompt.
> > > >
> > >
> > > This sounds like a natural (and likely, not too difficult) thing to do; Like the reviewer, I’d want to see these results.
> > >
> > > This is important since in various places **you make claims about length**: “our proposed model has the ability to generate variable length sequences in a non-autoregressive fashion, which is not possible with the Mask DFM construction.”
> > >
> > > > Edit Flows were able to train with 3x more data sequences per iteration because of their compute efficiency.
> > > >
> > >
> > > Just to be really sure: do you use 3x larger batch sizes?
> > >
> > > ---
> > >
> > > Other minor comments:
> > >
> > > - I +1 `arpi`'s comment that Figure 8 is very nice
> > >
> > > ---
> > >
> > > Looking forward to your response.

---

> > > > ### Author Response · Authors · 2025-08-06
> > > > **Reply**
> > > >
> > > > Thank you for the detailed reply to our rebuttal. We appreciate your feedback in helping us improve our manuscript. We first address the high level concerns, then answer each of you questions below:
> > > >
> > > > ## High level concerns
> > > >
> > > >
> > > > > The empirical gains:
> > > > > Are the significant?
> > > > > Are they consistent across different models?
> > > > > Do they generalize to longer sequences?
> > > >
> > > > We tested 3 domains, 2 architectures and 18 benchmarks in total. Edit Flows outperformed the non-autoregressive baseline (Mask DFM which is the most common model right now) on all of them.
> > > >
> > > > The paper demonstrates that the method works well at context length 128 for image captioning and at context length 1024 for code and text generation. These context length are appropriate for the model size and the tasks at hand. Analogous to AR models, scaling beyond this context length is merely a question of model size and computational resources.
> > > >
> > > > > The conceptual understand: what you adding to the literature?
> > > >
> > > > We re-iterate our modeling contributions:
> > > > 1. We introduce a non-autoregressive generation framework expanding upon the Discrete Flow Matching recipe, with native support for variable-length generation by introducing insertions and deletions.
> > > > 2. We construct a sequence-level probability path, enabling CTMC-based modeling directly over sequences of varying lengths, unlike prior work focused on token-level transitions.
> > > >
> > > > ## Specific questions
> > > >
> > > > > Then how does the model know what task you want it to do? Ultimately, you need some sort of lever to steer the systems. What is that “lever”? This makes me wonder if the reported results are purely doe to random chance. For your results in table 2,3, can you please provide their confidence intervals with Bootstrap resampling?
> > > >
> > > > During training, we assign randomly assign a portion of the sequence as conditioning (i.e. prompt).
> > > >
> > > > As a result, Edit Flows have the same levers as GPT models such as GPT-2 or Llama3. Given a prompt, they assign likelihoods for individual answers in multiple choice tasks or sample likely continuations in generation tasks. Both of these are stochastic, so there is some variance. We can add the confidence intervals to Tables 2 and 3, we are confident that the results are not due to chance.
> > > >
> > > > > I see why you’re avoiding “few-shot” setup: you’re starting from noise/null sequence and hence, it does seem (to my understanding) But by doing so, you do not give your model any ability be controlled. While you may consider “few-shot” eval unfair, others may consider it a fair game: first, coming up with 3-4 examples it not a big deal (I can do it in 10 mins). Plus, auto-regressive models have out-of-box in-context learning (you don’t need to train them for it).
> > > >
> > > > Edit Flows are able to do few-shot classification the same way autoregressive models can: by including the examples in the prompt. We do not consider this unfair, it is simply not the setting that we tested in our experiments.
> > > >
> > > > > I am going to re-iterate this: “OBQA and ARC-C (Table 2), the numbers are barely above random (25%).” Why should I believe that your numbers/gains any meaningful?
> > > >
> > > > These numbers are meaningfully different from 25%. Edit Flows reach 37% on OBQA, a benchmark that contains 500 questions. A random model has less than 1 in $10^8$ chance of reaching this or a better result. On ARC-C, Edit Flows reach 34% out of 1165 questions. A random model has less than 1 in $10^{11}$ chance of obtaining this or a better result.
> > > >
> > > > All of our experiments are comparing differences only in the modeling, removing other confounding variables as best as possible. As such, all baselines are trained on the same dataset with the same number of training FLOPS. This provides a meaningful comparison at a select parameter count and token count. Moreover, the same model is used for each task, so despite some low scoring values in some tasks, we clearly see that all models are significantly different than random sampling.
> > > >
> > > > > When I look at the sampled generations in Figure 8, there is a sense in which the model starts with generating most important/informative tokens (e.g, “black” for the picture of the dog with a black hat on). Alternatively, perhaps the examples are cherry-picked?
> > > >
> > > > No, the model has its own confidence in predicting certain tokens. When the model predicts an insertion, it also predicts the token to be inserted, and while the training signal is completely uniform, the model's predictions are usually far from uniform. We find the model tends to prioritize relevant tokens or completing half-words.

---

> > > > > ### Author Response · Authors · 2025-08-06
> > > > > **Reply Part 2**
> > > > >
> > > > > > Can you point me to where this is discussed in your text?
> > > > >
> > > > > Our model being a continiuous-time Markov process is discussed in Sections 2 and 3, and the framework of continuous-time Markov chains are used in most continous-time discrete diffusion or flow matching papers which we've referenced. From Eq 1, we define a terminating Markov process from t=0 to t=1. Section 2.2 then discusses how to build a generative model that samples q(x1) by interpolating between x0 and x1 samples, and shows in Eq. 7 that there exists a CTMC model that can sample from the data distribution q(x1). Section 3 then discusses how we build a CTMC that transports between sequences, by modeling rates (Eq. 13-15) that are decomposed into (for instance Eq. 13) probability to perform an insertion operation and what token to insert. *Conversely, when the model outputs zero probability for insertions, the model stops inserting.* The discussion of how this expands additional capabilities into existing models is then discusssed, in the paragraph on line 104. Given our discussion with the reviewer, we will include more emphasis on interpreting and understanding these equations, in particular, a paragraph before line 104; however, we note the mathematical constructions in this paper are fully self-contained and we provide many references for those unfamiliar with CTMC in these preliminary sections.
> > > > >
> > > > > > But the separation of roles in this reference is a shallow one and not a crucial aspect of the work, to my limited understanding. The main thesis here is casting generation as an iterative refinement, akin to yours. [...] I do think that the references [2,4] are also quite similar to some of your key contributions in terms of incremental text editing/addition/deletion.
> > > > >
> > > > > We are thankful to the reviewer for bringing our attention to these works: we agree that they are analogous to Edit Flows in that they use incremental edits to refine the target sequence. Hence we have agreed to include them in our related works section. However, we emhasize that there are key differences between the referenced works and Edit Flows that we detailed in our previous response.
> > > > >
> > > > >  - CGMH [2] is a sampling algorithm and is designed to be used with a pretrained autoregressive language model, whereas our proposed method is to train a model that directly proposes edit operations. [2] proposes using a forward and a backward autoregressive language model; and as a consequence, [2] *requires multiple model evaluations per step (same number as the number of proposed edits)* which is somewhat alleviated by a pre-selection (limiting the number of edits), whereas *our model by design uses only one evaluation to output all possible edits' probabilities*. [2] relies on the Metropolis-Hastings algorithm to run a stationary MCMC chain and requires accurately computing accept-reject ratios, whereas the CTMC framework we use does not require accept-rejects and is a non-stationary Markov process. [2]'s ultimate goal is to sample from autoregressive language models (treating the model as a target distribution), while our goal is to propose a completely different model (both training and sampling) and treating the data distribution as the target.
> > > > >
> > > > >   - G2LC [4] is also a sampling algorithm, designed to be used with a pretrained autoregressive language model. [3] differs from [2] in using treating it as an optimization problem instead of sampling, and uses the gradients of an LLM to (heuristically) determine edit operations; this gets around the efficiency problem of [2] but has no theoretical guarantees and cannot prove what distribution they are sampling from.
> > > > >
> > > > > > Can you please share a few examples of your training data?
> > > > >
> > > > > The datasets for text and code are available on HuggingFace. We are not allowed to include links in our reply, but a quick web search for `dclm-baseline-1.0` and `bigcode/the-stack` should lead to the right training sets.
> > > > >
> > > > > The training examples are quite long, so here we only include partial sequences.
> > > > >
> > > > > DCLM-baseline 1.0 example:
> > > > > ```
> > > > > IR Atmospheric Windows
> > > > >
> > > > > The Universe sends us light at all wavelengths of the electromagnetic spectrum. However, most of this light does not reach us at ground level here on Earth. Why? Because we have an atmosphere which blocks out many types of radiation while letting other types through.
> > > > > ...
> > > > > ```
> > > > >
> > > > > The stack example:
> > > > > ```
> > > > > class ZCL_IM__GTT_SOF_LE_SHIPMNT definition
> > > > > public
> > > > > final
> > > > > create public .
> > > > >
> > > > > public section.
> > > > >
> > > > > interfaces IF_EX_BADI_LE_SHIPMENT .
> > > > > protected section.
> > > > > private section.
> > > > > ENDCLASS.
> > > > > ...
> > > > > ```
> > > > >
> > > > >
> > > > > > Can you please share a few lines of your code pertain to your training objective? (I see the equations but I’d like to see how you’ve implemented it)
> > > > >
> > > > > Below we share a code-snippet free of proprietary dependencies.

---

> > > > > > ### Author Response · Authors · 2025-08-06
> > > > > > **Reply Part 3**
> > > > > >
> > > > > > > The reviewer is also saying this too: “their own when trained from scratch”. They’re not [necessarily] suggesting that you give additional information to the model. But rather, the reviewer’s concern is whether your gains are limited to weak models.
> > > > > >
> > > > > > The experiments in the paper serve as an apples-to-apples comparison at 280M scale. This supports the argument that our method is applicable to image captioning and it improves results at this scale. We believe that 280M parameter models are reasonable benchmarks for a methodology paper. While not SotA, these models are powerful and useful. They strike a balance between performance and not being too costly to train from scratch.
> > > > > >
> > > > > > > Well, this is clearly an unfair comparison to contrast llama3 vs GPT-2. You’d need to repeat the whole table for llama3 for apples to apples comparison. The reviewer is asking you to repeat the same table, for a fixed model (for difference choices of models).
> > > > > >
> > > > > > While not apples-to-apples comparison, it shows that Edit Flows are able to outperform AR models at a similar parameter count. We intended these results only to answer the reviewers' question regarding comparison to other references and it will not be included in the manuscript.
> > > > > >
> > > > > > > This sounds like a natural (and likely, not too difficult) thing to do; Like the reviewer, I’d want to see these results. This is important since in various places you make claims about length: “our proposed model has the ability to generate variable length sequences in a non-autoregressive fashion, which is not possible with the Mask DFM construction.”
> > > > > >
> > > > > > To clarify, we claim *our model can generate variable length sequences of up to 1024.* This number is just the maximum length that our models were trained on during training, and is shared with all baselines. Both Edit Flows and AR models do not perform well when generating a sequence longer than what they encounter during training.
> > > > > >
> > > > > > Thank you for bringing our attention to this: we understand that our statement regarding variable length generation may give the impression to the reader that Edit Flows address the context-window limitation of AR models. This is not the case. Following the reviewer's feedback, we clarify this limitation in the manuscript.
> > > > > >
> > > > > > > Just to be really sure: do you use 3x larger batch sizes?
> > > > > >
> > > > > > On average yes. This is to match the same compute (training FLOPS) as the baseline models. Edit flows use roughly 3x less compute per data sequence since we delete 2/3 of the tokens on average.  Flex Attention is then used to pack as many variable-length sequences as possible into a fixed total token budget, which is shared across all models.

---

> > > > > > > ### Author Response · Authors · 2025-08-06
> > > > > > > **Reply Part 4 (code example)**
> > > > > > >
> > > > > > > Below is a partial code snippet for demonstration purposes. To keep it simple and free of proprietary dependencies, we did not include the following features:
> > > > > > > * Batching
> > > > > > > * Conditioning on a random portion of the sequence
> > > > > > > * Scaling the model outputs by the rate
> > > > > > >
> > > > > > > ```
> > > > > > > [...]
> > > > > > >
> > > > > > > def get_z(ids: list[int]) -> tuple[list[int], list[int]]:
> > > > > > >     num_substitutions = len(ids) - max(len(ids) - target_num_substitutions, 0)
> > > > > > >     num_deletions = target_num_deletions + target_num_substitutions - num_substitutions
> > > > > > >     x_0 = [
> > > > > > >         int(token)
> > > > > > >         for token in np.random.randint(
> > > > > > >             low=0, high=vocab_size, size=[num_deletions + num_substitutions]
> > > > > > >         )
> > > > > > >     ]
> > > > > > >     sub_id = 0
> > > > > > >     z = (
> > > > > > >         [epsilon_0_id] * (len(ids) - num_substitutions)
> > > > > > >         + [epsilon_1_id] * num_deletions
> > > > > > >         + [sub_id] * num_substitutions
> > > > > > >     )
> > > > > > >     random.shuffle(z)
> > > > > > >
> > > > > > >     z_0: list[int] = []
> > > > > > >     z_1: list[int] = []
> > > > > > >     ids_index = 0
> > > > > > >     x_0_index = 0
> > > > > > >     for token in z:
> > > > > > >         if token == epsilon_1_id:
> > > > > > >             z_0.append(x_0[x_0_index])
> > > > > > >             z_1.append(epsilon_1_id)
> > > > > > >             x_0_index += 1
> > > > > > >         elif token == epsilon_0_id:
> > > > > > >             z_0.append(epsilon_0_id)
> > > > > > >             z_1.append(ids[ids_index])
> > > > > > >             ids_index += 1
> > > > > > >         elif token == sub_id:
> > > > > > >             z_0.append(x_0[x_0_index])
> > > > > > >             z_1.append(ids[ids_index])
> > > > > > >             x_0_index += 1
> > > > > > >             ids_index += 1
> > > > > > >     return z_0, z_1
> > > > > > >
> > > > > > >
> > > > > > > def get_z_t(z_0: list[int], z_1: list[int], kappa: float) -> list[int]:
> > > > > > >     return [
> > > > > > >         token_0 if np.random.uniform() > kappa else token_1
> > > > > > >         for token_0, token_1 in zip(z_0, z_1)
> > > > > > >     ]
> > > > > > >
> > > > > > > [...]
> > > > > > >     # Training loop
> > > > > > >     for sample in training_samples:
> > > > > > >         tokens: list[int] = encode(sample, bos=False)
> > > > > > >         z_0, z_1 = get_z(tokens)
> > > > > > >         z_0 = [bos_id] + z_0
> > > > > > >         z_1 = [bos_id] + z_1
> > > > > > >         t: float = np.random.uniform()
> > > > > > >         kappa: float = t  # Using a linear schedule
> > > > > > >         dkappa: float = 1.0
> > > > > > >         z_t: list[int] = get_z_t(z_0, z_1, kappa)
> > > > > > >         x_t: list[int] = remove_epsilon(z_t)
> > > > > > >         x_t_tensor: torch.Tensor = torch.tensor(x_t).to(device)
> > > > > > >
> > > > > > >         # Forward pass
> > > > > > >         insert_lambda, insert_q, delete_lambda, substitute_lambda, substitute_q = model(
> > > > > > >             x_t_tensor, t
> > > > > > >         )
> > > > > > >
> > > > > > >         # Calculate loss
> > > > > > >         loss_term_1: torch.Tensor = torch.sum(
> > > > > > >             insert_lambda + delete_lambda + substitute_lambda
> > > > > > >         )
> > > > > > >         loss_term_2: torch.Tensor = torch.tensor(0.0, device=device)
> > > > > > >         x_t_index: int = -1  # Corresponding index in x_t
> > > > > > >         for token_t, token_1 in zip(z_t, z_1):
> > > > > > >             if token_t != epsilon_0_id and token_t != epsilon_1_id:
> > > > > > >                 x_t_index += 1
> > > > > > >
> > > > > > >             if token_t == epsilon_0_id and token_1 != epsilon_1_id:
> > > > > > >                 # Missing token must be inserted
> > > > > > >                 loss_term_2 = loss_term_2 - (dkappa / (1 - kappa)) * torch.log(
> > > > > > >                     insert_lambda[x_t_index] * insert_q[x_t_index, token_1]
> > > > > > >                 )
> > > > > > >             elif token_t != epsilon_0_id and token_1 == epsilon_1_id:
> > > > > > >                 # Extra token must be deleted
> > > > > > >                 loss_term_2 = loss_term_2 - (dkappa / (1 - kappa)) * torch.log(
> > > > > > >                     delete_lambda[x_t_index]
> > > > > > >                 )
> > > > > > >             elif (
> > > > > > >                 token_t != epsilon_0_id
> > > > > > >                 and token_1 != epsilon_1_id
> > > > > > >                 and token_t != token_1
> > > > > > >             ):
> > > > > > >                 # Incorrect token must be substituted
> > > > > > >                 loss_term_2 = loss_term_2 - (dkappa / (1 - kappa)) * torch.log(
> > > > > > >                     substitute_lambda[x_t_index] * substitute_q[x_t_index]
> > > > > > >                 )
> > > > > > >
> > > > > > >         loss: torch.Tensor = loss_term_1 + loss_term_2
> > > > > > >         optimizer.zero_grad()
> > > > > > >         loss.backward()
> > > > > > >         optimizer.step()
> > > > > > > ```

---

> > > > > > > > ### Comment · Reviewer_czyg · 2025-08-07
> > > > > > > >
> > > > > > > > > We can add the confidence intervals to Tables 2 and 3, we are confident that the results are not due to chance.
> > > > > > > > >
> > > > > > > >
> > > > > > > > Yes, please share the results.
> > > > > > > >
> > > > > > > > > We do not consider this unfair, it is simply not the setting that we tested in our experiments.
> > > > > > > > >
> > > > > > > >
> > > > > > > > Great! If it’s not difficult to reproduce, I’d love to see the results. Since you already have your models trained, running it shouldn’t be too difficult, I believe.
> > > > > > > >
> > > > > > > > > The training examples are quite long, so here we only include partial sequences.
> > > > > > > > >
> > > > > > > >
> > > > > > > > Sorry or not being clear. I am familiar with these pre-training datasets. My question was examples of your data after your pre-processing the data. Basically, the data that goes inside your model for training how does it look? (input and the target) I am guessing it’d look sth like, partial sentences as input (words deleted) and the target would be slight (1-2 word) reconstructed versions?
> > > > > > > >
> > > > > > > > >
> > > > > > > > >
> > > > > > > > >
> > > > > > > > > > This sounds like a natural (and likely, not too difficult) thing to do; Like the reviewer, I’d want to see these results. This is important since in various places you make claims about length: “our proposed model has the ability to generate variable length sequences in a non-autoregressive fashion, which is not possible with the Mask DFM construction.”
> > > > > > > > > >
> > > > > > > > >
> > > > > > > > > To clarify, we claim *our model can generate variable length sequences of up to 1024.* This number is just the maximum length that our models were trained on during training, and is shared with all baselines. Both Edit Flows and AR models do not perform well when generating a sequence longer than what they encounter during training.
> > > > > > > > >
> > > > > > > >
> > > > > > > > Right. My point is that, perhaps you should explicitly acknowledge this as a weakness?
> > > > > > > >
> > > > > > > > Also, again, like the reviewer, I’d want to see these results on longer context generalization.
> > > > > > > >
> > > > > > > >  Thanks for answering the rest of the questions.

---

> > > > > > > > > ### Author Response · Authors · 2025-08-09
> > > > > > > > > **Reply**
> > > > > > > > >
> > > > > > > > > > We can add the confidence intervals to Tables 2 and 3, we are confident that the results are not due to chance.
> > > > > > > > > > Yes, please share the results.
> > > > > > > > >
> > > > > > > > > We provide the confidence interval for the mean calculated based on 10 evaluation runs for each benchmark. Because we only had a short time to comply with the request and since the reviewer has mainly been concerned with the choice tasks, we prioritized computing the confidence intervals for Table 2. For Table 3, based on a small number of previous runs, we find that there is similarly little deviation (<1% which is one or two test problems) across random seeds. Note the AR baseline is deterministic here, so no confidence interval is needed.
> > > > > > > > >
> > > > > > > > > Edit Flows (CFG applied to $\mathcal{L}$):
> > > > > > > > >
> > > > > > > > > Dataset       | Mean   | 95% CI for the mean
> > > > > > > > > --------------|--------|--------------------
> > > > > > > > > HellaSWAG     | 56.96% | [56.71, 57.21]
> > > > > > > > > ARC Easy      | 61.25% | [60.60, 61.90]
> > > > > > > > > ARC Challenge | 34.15% | [33.68, 34.62]
> > > > > > > > > PIQA          | 67.01% | [66.39, 67.63]
> > > > > > > > > OBQA          | 35.58% | [35.07, 36.09]
> > > > > > > > > Winogrande    | 53.19% | [52.84, 53.54]
> > > > > > > > >
> > > > > > > > > This shows that our results are not due to random chance.
> > > > > > > > >
> > > > > > > > > > Great! If it’s not difficult to reproduce, I’d love to see the results. Since you already have your models trained, running it shouldn’t be too difficult, I believe.
> > > > > > > > >
> > > > > > > > > We ran the benchmarks in the 3-shot setting with Edit Flows (CFG applied to $\mathcal{L}$) and the AR baseline.
> > > > > > > > >
> > > > > > > > > Dataset     | Edit Flows | AR Baseline
> > > > > > > > > ------------|------------|------------
> > > > > > > > > Hellaswag   | 56.4%      | 49.5%
> > > > > > > > > ARC-E       | 61.8%      | 72.6%
> > > > > > > > > ARC-C       | 31.3%      | 37.4%
> > > > > > > > > PIQA        | 66.1%      | 75.8%
> > > > > > > > > OBQA        | 36.2%      | 33.0%
> > > > > > > > > WinoGrande  | 54.2%      | 61.9%
> > > > > > > > >
> > > > > > > > > As can be seen, though some values did go beyond the 95% CI, overall a minor difference.

---

> > > > > > > > > > ### Author Response · Authors · 2025-08-09
> > > > > > > > > > **Reply part 2**
> > > > > > > > > >
> > > > > > > > > > > My question was examples of your data after your pre-processing the data. Basically, the data that goes inside your model for training how does it look? (input and the target) I am guessing it’d look sth like, partial sentences as input (words deleted) and the target would be slight (1-2 word) reconstructed versions?
> > > > > > > > > >
> > > > > > > > > > We showcase the training examples in the image captioning task, because they are short and therefore, they make for good demonstration. The training data are pairs of images (used as conditioning) and captions. From the caption, we programmatically generate Z_0, Z_1, Z_t and X_t: We first sample Z_0 (the noisy sequence) and align it with Z_1 (random alignment). Then sample t between 0 and 1, and finally Z_t. X_t is simply Z_t without `<EPS>` tokens.
> > > > > > > > > >
> > > > > > > > > > In the code example, the model takes X_t and t as inputs where it is marked `# Forward pass`. At `# Calculate loss`, the two loss terms are computed with respect to the target X_1 (Figure 3 and Eq 23). The model is trained to predict *all* the edits needed to obtain X_1 from X_t. This is usually more than 1-2 edits.
> > > > > > > > > >
> > > > > > > > > > Examples of the probability path where X_0 is the empty string.
> > > > > > > > > > ```
> > > > > > > > > > Training sample X_1: person and her sister at the graduation of their brother
> > > > > > > > > >   Z_0: <IMG> <BOS> <EPS> <EPS> <EPS> <EPS> <EPS> <EPS> <EPS> <EPS> <EPS> <EPS>
> > > > > > > > > >   Z_1: <IMG> <BOS> person and her sister at the graduation of their brother
> > > > > > > > > >   t:   0.60
> > > > > > > > > >   Z_t: <IMG> <BOS> person <EPS> <EPS> <EPS> <EPS> <EPS> <EPS> of <EPS> <EPS>
> > > > > > > > > >   X_t: <IMG> <BOS> person of
> > > > > > > > > >
> > > > > > > > > > Training sample X_1: mix all the ingredients in a big bowl
> > > > > > > > > >   Z_0: <IMG> <BOS> <EPS> <EPS> <EPS> <EPS> <EPS> <EPS> <EPS> <EPS>
> > > > > > > > > >   Z_1: <IMG> <BOS> mix all the ingredients in a big bowl
> > > > > > > > > >   t:   0.82
> > > > > > > > > >   Z_t: <IMG> <BOS> mix <EPS> <EPS> ingredients <EPS> a <EPS> bowl
> > > > > > > > > >   X_t: <IMG> <BOS> mix ingredients a bowl
> > > > > > > > > >
> > > > > > > > > > Training sample X_1: worried couple sitting at the table with empty wallets
> > > > > > > > > >   Z_0: <IMG> <BOS> <EPS> <EPS> <EPS> <EPS> <EPS> <EPS> <EPS> <EPS> <EPS>
> > > > > > > > > >   Z_1: <IMG> <BOS> worried couple sitting at the table with empty wallets
> > > > > > > > > >   t:   0.30
> > > > > > > > > >   Z_t: <IMG> <BOS> <EPS> <EPS> <EPS> <EPS> <EPS> <EPS> <EPS> <EPS> <EPS>
> > > > > > > > > >   X_t: <IMG> <BOS>
> > > > > > > > > > ```
> > > > > > > > > >
> > > > > > > > > > Examples of the probability path where X_0 has 4 uniform random tokens. 2 to be deleted and 2 to be substituted.
> > > > > > > > > > ```
> > > > > > > > > > Training sample X_1: happy birthday sign on a chalkboard at party
> > > > > > > > > >   Z_0: <IMG> <BOS> <EPS> lounge <EPS> <EPS> the person <EPS> <EPS> <EPS> birthday
> > > > > > > > > >   Z_1: <IMG> <BOS> happy birthday sign on <EPS> <EPS> a chalkboard at party
> > > > > > > > > >   t:   0.26
> > > > > > > > > >   Z_t: <IMG> <BOS> <EPS> lounge <EPS> <EPS> the person <EPS> <EPS> <EPS> birthday
> > > > > > > > > >   X_t: <IMG> <BOS> lounge the person birthday
> > > > > > > > > >
> > > > > > > > > > Training sample X_1: view of deck from the driveway
> > > > > > > > > >   Z_0: <IMG> <BOS> beach <EPS> driveway person ridge <EPS> <EPS> <EPS>
> > > > > > > > > >   Z_1: <IMG> <BOS> <EPS> view of deck <EPS> from the driveway
> > > > > > > > > >   t:   0.16
> > > > > > > > > >   Z_t: <IMG> <BOS> beach <EPS> driveway person ridge <EPS> <EPS> <EPS>
> > > > > > > > > >   X_t: <IMG> <BOS> beach driveway person ridge
> > > > > > > > > >
> > > > > > > > > > Training sample X_1: portrait of a young woman
> > > > > > > > > >   Z_0: <IMG> <BOS> bowl <EPS> above big <EPS> party <EPS>
> > > > > > > > > >   Z_1: <IMG> <BOS> portrait of <EPS> a young <EPS> woman
> > > > > > > > > >   t:   0.95
> > > > > > > > > >   Z_t: <IMG> <BOS> portrait of <EPS> a young <EPS> woman
> > > > > > > > > >   X_t: <IMG> <BOS> portrait of a young woman
> > > > > > > > > > ```
> > > > > > > > > >
> > > > > > > > > > > To clarify, we claim our model can generate variable length sequences of up to 1024. This number is just the maximum length that our models were trained on during training, and is shared with all baselines. Both Edit Flows and AR models do not perform well when generating a sequence longer than what they encounter during training.
> > > > > > > > > > > Right. My point is that, perhaps you should explicitly acknowledge this as a weakness?
> > > > > > > > > >
> > > > > > > > > > Yes, we agree and will make it clear that we are not addressing the sequence length generalization capabilities of language models. The total context length and generation lengths do not exceed 1024 for all the models, including baselines, that we trained. This is a completely orthogonal research problem to what we are tackling.
> > > > > > > > > >
> > > > > > > > > > > Also, again, like the reviewer, I’d want to see these results on longer context generalization.
> > > > > > > > > >
> > > > > > > > > > As we shared in our response to reviewer C6NG, for problems with longer contexts, a basic sliding window is performed: if a prompt was longer than 768 tokens, we only used the last 768 tokens of the prompt as conditioning, leaving the rest of the tokens for generation. As discussed above, generalizing beyond the training context length is orthogonal and outside the scope of this paper.
> > > > > > > > > >
> > > > > > > > > > In previous replies, we answered the reviewer's questions regarding differences to prior work and explained the core methodology in detail, and with this latest reply confirmed that our results are not due to random chance. We hope this clarifies the points under discussion. We kindly ask the reviewer to consider that we have thoroughly responded to their concerns regarding the novelty of the method, distinctions from prior work, and the statistical significance of the empirical results.

---

> ### Author Response · Authors · 2025-08-09
>
> > In your 3-shot experiment (which, in my view, uses too few shots), the AR baseline shows a larger improvement over its 0-shot counterpart than your Edit Flows approach does (3-shot vs. 0-shot).
>
> We emphasize that both the AR and Edit Flow numbers are close to the 0-shot performance. The few-shot setting does not appear to yield a significant improvement for any of the methods presented here.
>
> > Your reported confidence intervals also appear problematic. For example, your CI for ARC-C is 0.47, which is surprisingly narrow for a test set of ~1,500 questions. It seems you have treated evaluations over 10 runs as independent trials, effectively computing the CI as if you had a dataset of size 10×1,500. This is incorrect, as multiple runs on the same questions are not independent and will be correlated. Using this logic, one could reduce CIs arbitrarily by increasing the number of runs, which does not make sense!! A more appropriate approach would be to perform the evaluation once and use bootstrap resampling to compute the CIs.
>
> We appreciate the reviewer’s attention to the calculation of confidence intervals and the potential misunderstandings arising from it. To clarify, the benchmark under consideration (e.g., ARC-C) consists of a fixed set of approximately 1,500 questions. _Following standard practice, our measurements are made on this fixed benchmark, not on a broader distribution of possible questions._ Thus, the metric we report (e.g., 34.15% on ARC-C) specifically denotes the expected performance on this particular set of questions, rather than the expected performance over the general task distribution.
>
> Given that Edit Flows are inherently stochastic, each evaluation run on the fixed benchmark produces a noisy outcome. To address this, we conduct 10 independent runs and compute the confidence interval for the expected performance on the benchmark using the central limit theorem: $\bar{x} \pm z^{95} \frac{\sigma}{\sqrt{n}}$. This approach provides an estimate for the expected score on the given benchmark, where $n$ is the number of runs. While this method is not perfect since we only use 10 trials, as the reviewer noted, the confidence intervals are tight. Therefore, we can be confident that our measurements are accurate and they are not due to random chance.
>
> We also appreciate the reviewer’s suggestion to use bootstrap resampling to compute confidence intervals as an alternative. Bootstrap resampling the set of questions would allow us to estimate the variability one would expect if a similar but slightly different set of questions were used for evaluation—thus providing an interval that more closely reflects uncertainty with respect to the overall data distribution. While this isn't a standard way for reporting evaluation metrics in the current community, this is indeed a valid and widely accepted method in statistical analysis and we will consider it in future work.
>
> We re-iterate that while we claim to outperform the mask model on all tasks, we do not claim outperforming the AR model. Furthermore, the reviewer's concerns have all been regarding multiple choice tasks, not generation tasks. Edit Flow's ability to generate variable length sequences is emphasized strongly in both the image captioning and code generation tasks, the two generation problems we report on.

---

### Official Review · Reviewer_LxY2 · 2025-06-30

**Clarity:** 3
**Significance:** 3
**Originality:** 3
**Rating:** 4
**Confidence:** 3

**Summary:**

The  authors introduces Edit Flows, a non-autoregressive generative framework that models sequence generation via edit operations (insertions, deletions, substitutions) within a Continuous-Time Markov Chain (CTMC). Unlike prior non-autoregressive models that rely on fixed-length token-wise transitions, Edit Flows naturally support variable-length generation by defining a discrete flow over sequences. The authors propose a tractable training method using auxiliary alignment processes and demonstrate strong empirical performance across image captioning, code generation, and open-ended text tasks.

**Questions:**

1.  Will the authors provide a minimal implementation free of proprietary dependencies?

2. Why does Edit Flows require 5K–10K sampling steps for code generation? Are there potential optimizations to reduce this computational cost without sacrificing output quality?

3. Why does Edit Flows still lag behind autoregressive models in text/code tasks? Is this a fundamental limitation of the approach, or could it be addressed with improved training or data?

**Ethical Concerns:**

["NO or VERY MINOR ethics concerns only"]

**Final Justification:**

The concerns have been addressed. Thus, I keep my rating.

**Limitations:**

yes

**Paper Formatting Concerns:**

no formatting concerns

**Quality:**

3

**Strengths And Weaknesses:**

Strengths
- The proposed method demonstrates strong novelty, and a solid theoretical foundation is provided.

- Extensive experimental evaluations are conducted on code generation and image captioning tasks, with promising results.

Weaknesses
- The performance of the proposed method still falls short of autoregressive approaches in the image captioning task.

- The method appears to require sampling 5,000–10,000 steps for code generation, which may raise latency concerns.

---

> ### Author Rebuttal · Authors · 2025-07-30
>
> Thank you for your detailed review and for highlighting both the strengths and areas for improvement in our paper. We appreciate the opportunity to address your questions and concerns. Here are our responses:
>
> > The performance of the proposed method still falls short of autoregressive approaches in the image captioning task.
>
> We note that the primary value of all of our experiments is to analyze the ability of the models on their own when trained from scratch, without additional data or pretrained weights. This allows us to analyze model performance differences without confounding variables such as varying training data. For instance, most papers that report image captioning use pretrained weights (e.g. VLP and ClipCap which we included for reference). When doing this apples-to-apples comparison, we outperform the autoregressive model.
>
> However, to satisfy our and the reviewer's curiosity, we have conducted an additional experiment where we initialize the weights using a Llama 3 1B model, as a means to test how well Edit Flows can adopt the weights of a pretrained AR model trained on internet-scale data. This increased our CIDEr score on MSCOCO from 108 to 124, which now outperforms both VLP and ClipCap (a 1.5B GPT-2 model). Going further than this would require additional ideas from the literature, such as data synthesis methods like recaptioning, which are orthogonal to the modeling framework and is out of scope for us.
>
> > Will the authors provide a minimal implementation free of proprietary dependencies?
>
> We are adding a code example for Edit Flows in the appendix. This example is designed to be minimal and free of proprietary dependencies, allowing for easy replication and further exploration by the community.
>
> > Why does Edit Flows require 5K–10K sampling steps for code generation? Are there potential optimizations to reduce this computational cost without sacrificing output quality?
>
> We decoupled the investigations into performance and sampling efficiency. In this work, we only optimized for generation quality.
>
> On the other hand, our sampling code is very minimalistic and designed for easy batching. A more sophisticated implementation would likely bound the number of model evaluations by the total number of edit operations per generated sequence. We believe that there is a large scope for improving generation speed. Most of the 5,000–10,000 steps don’t change the input at all (and we informally found that only approximately 10-20% of the steps have an edit operation performed on the state). We hope to reduce the significant inference-time computational cost of Edit Flows in future work.
>
> > Why does Edit Flows still lag behind autoregressive models in text/code tasks? Is this a fundamental limitation of the approach, or could it be addressed with improved training or data?
>
> Our experiments show that Edit Flows outperform the mask construction, the current most popular non-AR model, in code generation. In our experiments, we do find that autoregressive models outperform Flow models on code benchmarks at this scale at equal compute cost. We theorize that Autoregressive models have the advantage that, due to their causal nature, they learn all possible left-conditioned sequences, whereas non-autoregressive models do not always condition on tokens that correspond to the prompt at inference time. Furthermore, we only used pretraining data, which contains large documents and a lot of sequences are larger than 1024 sequence length which we had to crop. We note that [1] showed recently that diffusion models are able to close this gap with further training and even outperform autoregressive models in data-limited settings. Though their data sets are much smaller than our setup, we believe future improvements in data efficiency can significant close this gap.
>
> We hope these responses address your concerns and provide clarity on the points raised. Thank you again for your valuable feedback.
>
> [1] Prabhudesai, Mihir, et al. "Diffusion Beats Autoregressive in Data-Constrained Settings." arXiv preprint arXiv:2507.15857 (2025).
>
> List of manuscript changes in response to the feedback:
> 1. Added a reference implementation of Edit Flows to the appendix.

---

> > ### Comment · Reviewer_LxY2 · 2025-08-05
> > **response**
> >
> > Thanks for the detailed response. And I have no more queries at this time.

---

### Official Review · Reviewer_C6NG · 2025-07-02

**Clarity:** 4
**Significance:** 4
**Originality:** 3
**Rating:** 6
**Confidence:** 4

**Summary:**

The paper develops a discrete flow framework with edit operations for the markov chain transitions instead of the masking (insertion/replacement). It constructs the corresponding theoretical framework, develops a high-performing practical implementation and designs a set of improved training and inference techniques (related to corruption strategies, classifier-free guidance, etc.). It evaluates the method on VQA, text and code generation and shows competitive performance with autoregressive models and better performance compared to masked diffusion methods.

**Questions:**

- Why do you think the model works worse for code generation?
- Does the model generalize well to the autoregressive generation order? DiffuCoder has recently shown that discrete text diffusion models somewhat diverge to the autoregressive order anyway. I was curious if something like that happens for your model and if not, how well would it generate autoregressively?
- Autoregressive models can generalize to longer in the sliding window fashion with overlaps. How well would the proposed model do the same?
- I suspect that δxt(x¬i) is a bit non-strictly defined in its current form since there are many inputs $x$ that would have a density of "infinity", so instead it should be defined as a mixture of delta distributions (a delta distribution per each "correct" input $x$).
- Did you perform any quantitative exploration of self-correction? There is many possible inference schedules for self-correction (i.e., how many steps to do, at which time-steps to do it, etc.). Have you run any evaluations for it?
- L300: "In our experiments, Edit Flows are able ingest 3× more training data per iteration while using the same compute and memory as Mask DFM.". Do you use 3x larger batch size or 3x longer (on average) sequence length to train Edit Flows?

**Ethical Concerns:**

["NO or VERY MINOR ethics concerns only"]

**Final Justification:**

I've read the authors' response and believe that it has resolved my concerns. I believe it's a very interesting paper and decided to increase my score.

I also went through the fellow negative review of czyg and believe that their raised concerns are nonsensical.

**Limitations:**

There is a reasonable discussion of the limitations in the experiments section.

**Paper Formatting Concerns:**

Figure 3 text is a bit too small, even though there is quite a lot of free space around

**Quality:**

4

**Strengths And Weaknesses:**

Strengths:
- The overall framework is novel and feels very natural for text generation tasks — i.e. much more natural than existing masked formulations that feel far-fetched. It's cool to see the out-of-the-box self-correction mechanism.
- The work develops a set of improved techniques for both training and inference, which makes it readily competitive with modern masked/autoregressive approaches
- The theoretical framework is rigorous and i didn't spot any problems with it (I have not gone through the proofs, but closely read through the exposition)
- The empirical results are thorough and convincing

Weaknesses:
- I'm not sure that it's fair to use 3x time more tokens to train Edit Flows compared to the baselines. What is the performance at 2T tokens?
- Appendix section "B.1 Localized propagation path" seems to be crucial for good performance, but it's 2 full pages of technical text. I spent ~10 minutes trying to understand the high level of what's going on, but couldn't do that. I understand that it's some "local edits", but it would be good to add a paragraph describing the high level idea for those who wants to understand the intuition, but do not need to know the gritty details.

---

> ### Author Rebuttal · Authors · 2025-07-30
>
> Thank you for your thoughtful and detailed review of our paper. We appreciate your insights and the opportunity to address your questions and concerns. Below, we provide responses to each of your points:
>
> > I'm not sure that it's fair to use 3x time more tokens to train Edit Flows compared to the baselines. What is the performance at 2T tokens?
>
> We were unclear in discussing this point. The compute budget of Edit Flows matches that of the baseline autoregressive model and Mask DFM in our experiments. Edit flows use roughly 3x less compute per data sequence than Mask DFM due to not needing `<MASK>` tokens.
>
> A simple explanation for the 3x efficiency gain is, for the $\kappa_t = t^3$ scheduler, we approximately remove 2/3 of the tokens per sequence. The Mask DFM model replaces removed tokens with a special `<MASK>` token, whereas Edit Flow actually deletes the tokens. This allows Edit Flows to use 3x less compute during training.
>
> To understand the effect of token efficiency better, we performed additional experiments. When we consume the same number of data sequences as the Mask DFM, our model achieves 12.1% on HumanEval pass@1. When we use the same compute budget as the Mask DFM, our model achieves 12.8% on HumanEval pass@1. Both are better than the Mask DFM which is at 9.1%.
>
> > Appendix section "B.1 Localized propagation path" seems to be crucial for good performance, but it's 2 full pages of technical text.
>
> The standard path will delete tokens independently, leading to e.g.
>
> $X_1$ = "The quick brown fox jumps over the lazy dog"
>
> $X_t$ = "The `<DEL>` brown fox `<DEL>` over `<DEL>` lazy `<DEL>`"
>
> The localized path is used to incentivize retaining subsequences, and results in e.g
>
> $X_t$ = "`<DEL>` quick brown fox `<DEL>` `<DEL>` `<DEL>` lazy dog"
>
> We are adding a figure showing how the token removal process is affected by a localized path in the appendix, which can provide high-level insights. We believe that the reason localized paths work better is because it is more aligned with the model's sampling process. During sampling, the model tends to generate well-formed subsequences since it may be easier to predict nearby tokens than faraway tokens (a frequent example is generating a token that completes half a word for multi-token words).
>
> > Why do you think the model works worse for code generation?
>
> Our experiments show that Edit Flows outperform the mask construction, the current most popular non-AR model, in code generation. In our experiments, we do find that autoregressive models outperform Flow models on code benchmarks at this scale at equal compute cost. We theorize that Autoregressive models have the advantage that, due to their causal nature, they learn all possible left-conditioned sequences, whereas non-autoregressive models do not always condition on tokens that correspond to the prompt at inference time. Furthermore, we only used pretraining data, which contains large documents and a lot of sequences are larger than 1024 sequence length which we had to crop. We note that [1] showed recently that diffusion models are able to close this gap with further training and even outperform autoregressive models in data-limited settings. Though their data sets are much smaller than our setup, we believe future improvements in data efficiency can significant close this gap.
>
> > Does the model generalize well to the autoregressive generation order?
>
> Unlike mask models, which are equivalent to any-order models, Edit Flows cannot be post-hoc queried to generate in an "autoregressive" order. The Edit Flow model predicts a bag-of-tokens that includes all missing tokens, but does not predict the order in which they should be inserted. We also tried using high-confidence sampling for Edit Flow models; it is not autoregressive and it performed on par with regular sampling. We are including additional qualitative examples to showcase the generation process on code examples.
>
> > Autoregressive models can generalize to longer in the sliding window fashion with overlaps. How well would the proposed model do the same?
>
> We used sequences of size 1024 for training both autoregressive and non-autoregressive models. For MBPP, had to resort to a similar method, because the prompt + generation length can be longer than 1024. In order to evaluate on MBPP, if a prompt was longer than 768 tokens, we only used the last 768 tokens of the prompt as conditioning. Conversely, we did not attempt evaluating the model on longer than 1024 sequence length, though we imagine it would not perform well compared to shortening the prompt.
>
> > I suspect that δxt(x¬i) is a bit non-strictly defined in its current form since there are many inputs  that would have a density of "infinity", so instead it should be defined as a mixture of delta distributions (a delta distribution per each "correct" input )
>
> We used $\delta$ as shorthand for the Kronecker delta function which is an indicator function (it is either 1 or 0) rather than as a Dirac-delta distribution. We thank you for pointing this out and we will make this definition clear.
>
> > Did you perform any quantitative exploration of self-correction? There is many possible inference schedules for self-correction (i.e., how many steps to do, at which time-steps to do it, etc.). Have you run any evaluations for it?
>
> We swept the hyperparameters for the divergence-free component (which uses both the forward and reverse rates) and found that it improved the results by +2.2% for the EditFlow model and by +1.9% for the Uniform $X_0$ + Edit Flow model on the HumanEval+ pass@1 benchmark.
>
> Additionally, we are updating the appendix with qualitative examples that showcase how Edit Flows are able to identify and fix bugs in an incorrect implementation.
>
> > L300: "In our experiments, Edit Flows are able ingest 3× more training data per iteration while using the same compute and memory as Mask DFM.". Do you use 3x larger batch size or 3x longer (on average) sequence length to train Edit Flows?
>
> The compute budget, the number of iterations and sequence length matched in all models. Edit Flows were able to train with 3x more data sequences per iteration because of their compute efficiency. See our detailed explanation and additional ablation experiment in Question 1.
>
> > Minor formatting concern: Figure 3 text is too small.
>
> Thank you for pointing this out. We have adjusted this figure to increase the font size and make the figure more legible.
>
> We hope these responses address your concerns and provide clarity on the points raised. Thank you again for your valuable feedback.
>
> [1] Prabhudesai, Mihir, et al. "Diffusion Beats Autoregressive in Data-Constrained Settings." arXiv preprint arXiv:2507.15857 (2025).
>
> List of manuscript changes in response to the feedback:
> 1. Expanded the discussion on the reason why Edit Flows use 3x less compute per token during training.
> 2. Included a figure to explain the sampling process of localized path model.
> 3. Clarified the notation of $\delta$.
> 4. Included qualitative examples from the code generation model to showcase the generation process.
> 5. Added a qualitative example to the appendix to showcase how Edit Flows can fix an incorrect implementation of the `is_prime` function.

---

> > ### Comment · Reviewer_C6NG · 2025-08-05
> >
> > I am thankful to the authors for providing a detailed response to address my concerns and questions, especially regarding the  corruption process and sampling. I would also be curious to see the bug fixing qualitatives.
> >
> > As to 3x more tokens used, I understood this part correctly, but thank you for the extended elaboration. My concern/question is that the Edit Flow model has seen 3x more text sequences during training, which can be deemed unfair w.r.t. vanilla MDM (e.g., it did 3x more epochs). I.e. how would MDM perform if it is trained on 3x shorter sequences (and then possibly fine-tuned on longer ones for a bit; or maybe it's possible to improve its throughput in some other way to circumvent the need for fine-tuning)? Please don't treat it as a request for experiments — I'm just thinking out loud and was curious to discuss this point.

---

### Official Review · Reviewer_arpi · 2025-07-03

**Clarity:** 4
**Significance:** 4
**Originality:** 3
**Rating:** 6
**Confidence:** 4

**Summary:**

Edit Flows proposes a discrete flow matching approach to sequence modeling using edit operations which allows for variable length processing with the flow matching framework, rather than the standard "blockwise with padding" approach generally taken. This increases training efficiency, and leads to numerous downstream benefits, as well as operating in a highly interpretable space of actions. Applied to image captioning, text generation, and code generation, Edit Flows shows competitive performance to relevant baselines, aligning with the intuitions and mathematical derivations, which justify the approach as well as numerous variants of interest.

**Questions:**

Will the authors release code, or pseudo-code for the key methods of this work, in order to facilitate reproduction in open source?

In Table 6, what were the CFG values tuned on? Train, train and validation, or validation only?

On figure 7, do the authors have any comparison of parallelization versus sampling steps to fit a given computational budget, similar to "Large Language Monkeys: Scaling Inference Compute with Repeated Sampling"? For example, is it better to do pass @64, with smaller steps, or pass @10 with more steps?

Similar to the above, reverse rates were discussed in the paper, but I did not see any concrete study of result quality over compute with reverse rate enabled, versus more parallelism or increased number of steps - do the authors have any results on this?

One concern I have on this approach is vocabulary scaling beyond 32k. Though this method should scale moderately well, do the authors have any empirical or theoretical insights into how the compute cost grows as the vocabulary increases, compared to (say) standard Transformer models? Ideally this would be similar to Transformer scaling, and allow for larger scale experiments with sufficient large scale compute to train up to the 20-70B range, with larger vocabulary sizes to match common models in that range.

Do the authors have any qualitative examples of injecting invalid text (false facts, code vulnerability, typos, etc) during the sampling chain, and seeing if the model returns to edit and remove these? The space of human-in-the-sampling-chain seems possible given the model setup, depending on this behavior. Similarly, given it should be possible to mark a subsequence "not for editing" - does having some subsequence marked fixed change the following sampling behavior?

Any intuition on CFG on rates versus CFG on Bregman divergences and when to prefer one or the other?

**Ethical Concerns:**

["NO or VERY MINOR ethics concerns only"]

**Final Justification:**

The updated rebuttal and discussion really highlighted the strengths of the method and expanded the impact of the work. I would rate this paper "strong accept" in general, without hesitation given the updates. Accounting for scaling limitations, the rest is very thorough and well done - the response to every reviewer has greatly enhanced the understanding and clarity of the work. If I saw this paper in *any* major conference, or even as a spotlight talk, it would make sense to me given the additional focus on results, scaling, and integration into some larger experimentation of interest to many researchers.

**Limitations:**

Yes

**Quality:**

4

**Strengths And Weaknesses:**

Strengths:
Edit Flows thoroughly introduces the background, key motivation, and driving theory behind the approach. It covers many relevant background papers, and a number of variations on Edit Flow methods for ablating key mechanisms. Figure 8 is very nice, and would make a compelling video visualization. Overall, the paper was an enjoyable read, and had an excellent treatment of the material, background, and various application areas, along with a detailed appendix.

Weaknesses:
It would be nice to see some larger and more recent image captioning models, to compare best-on-best performance (though this may need to be independent of model size). As it stands, the qualitative examples in the appendix are compelling.

The "Autoregressive" comparison in the tables is a bit too generic, providing a relevant citation of which model, if it was self-trained or from another paper, and so on. The citations in section 5 indicate it is a Transformer, but what flavor? Being detailed in the tables will help contextualize the results. Reusing the same superscript, to mean different things ("not comparable" in one table, and "our own implementation" in another) is also something I would change.

Generally, as always, more comparisons would help, and especially pre-trained comparisons would be useful given the recent uptick in publicly available mask-based generators. There are a huge amount of comparisons that could be used in Table 1 or Table 2 even if some need qualification (different training set, different model size, and so on), and some qualitative study of failure cases could be really useful/

The efficiency argument in appendix D is a bit of a drawback in disguise - what do the baselines look like with equivalent token budgets (even if it might take 3x more steps, and more compute)? This may not be feasible, but some discussion about this point is worthwhile given the importance of training, overtraining, and matching token budgets in general. Alternatively, reporting Edit Flow performance at 2T tokens could highlight the same. Being more efficient in terms of tokens-per-step is definitely a benefit, but being careful in the competitive comparison is important.

The mathematical treatment here is dense, in line with other work in this area, but some condensing of the mathematical bits to a relevant section, wrapped by higher level descriptions of the key ideas *behind* the formulations, could help readers who are not familiar with the flow matching literature and notations get to the meat of the contribution.

---

> ### Author Rebuttal · Authors · 2025-07-30
>
> Thank you very much for your thorough and insightful review of our submission. We appreciate your detailed feedback and the time you took to engage with our work. Below, we address each of your questions and concerns in turn.
>
> > It would be nice to see some larger and more recent image captioning models, to compare best-on-best performance (though this may need to be independent of model size). As it stands, the qualitative examples in the appendix are compelling.
>
> We note that the primary value of all of our experiments is to analyze the ability of the models on their own when trained from scratch, without additional data or pretrained weights. This allows us to examine model performance differences without confounding variables such as varying training data. Most papers that report image captioning use pretrained weights (e.g. VLP and ClipCap which we included for reference).
>
> However, to satisfy our and the reviewer's curiosity, we have conducted an additional experiment where we initialize the weights using a Llama 3 1B model, as a means to test how well Edit Flows can adopt the weights of a pretrained AR model trained on internet-scale data. This increased our CIDEr score on MSCOCO from 108 to 124, which now outperforms both VLP and ClipCap (a 1.5B GPT-2 model). Going further than this would require additional ideas from the literature, such as data synthesis methods like recaptioning, which are orthogonal to the modeling framework and is out of scope for us.
>
> > The "Autoregressive" comparison in the tables is a bit too generic, providing a relevant citation of which model, if it was self-trained or from another paper, and so on. The citations in section 5 indicate it is a Transformer, but what flavor? Being detailed in the tables will help contextualize the results.
>
> Thank you for pointing this out. We use the 280M and 1.3B parameter variants of the Llama3 architecture based on the official Llama3 repository. We updated the tables in the manuscript to denote this clearly. Further architecture and training details are included in appendix D.
>
> > Reusing the same superscript, to mean different things ("not comparable" in one table, and "our own implementation" in another) is also something I would change.
>
> Thank you for raising this issue, as we did not notice this. We updated the superscripts to be consistent across all tables.
>
> > There are a huge amount of comparisons that could be used in Table 1 or Table 2 even if some need qualification (different training set, different model size, and so on), and some qualitative study of failure cases could be really useful/
>
> We understand the reviewer's desire to scale up, as we would like to as well. However, we emphasize that the value of our experiments is in having a concrete apples-to-apples comparison on the same data and compute scale. This is crucial for understanding design choices, and it is very difficult to understand the tradeoffs of different modeling approaches when there are confounding variables such as data and archiecture differences. We will investigate scaling laws in a separate work, which will provide clearer insights for comparing across different model sizes than a tabular comparison.
>
> For a qualitative comparison, we generated outputs from the Mask DFM and Edit Flow models on the code benchmarks and we are includin them in the appendix. Similar to prior work, we find that when mask models are trained for variable length generation using <PAD> tokens, their predictions over-emphasize the <PAD> tokens, and thus confidence-based unmasking fails as it prioritizes sampling <PAD> tokens first. In contrast, Edit Flow models never see padding tokens, though we find that regular sampling and confidence-based sampling perform on par.
>
> > The efficiency argument in appendix D is a bit of a drawback in disguise - what do the baselines look like with equivalent token budgets (even if it might take 3x more steps, and more compute)? This may not be feasible, but some discussion about this point is worthwhile given the importance of training, overtraining, and matching token budgets in general.
>
> We evaluated the models in the equal compute setting because we believe this is the most fair comparison. Had we shown a more modest improvement while using 3x less compute, we would be understating the benefits of our approach.
>
> To understand the effect of token efficiency better, we performed additional experiments on the code benchmarks. When we consume the same number of data sequences as the Mask DFM, our model achieves 12.1% on HumanEval pass@1. When we use the same compute budget as the Mask DFM, our model achieves 12.8% on HumanEval pass@1. Both are better than the Mask DFM which is at 9.1%.
>
>
> > Will the authors release code, or pseudo-code for the key methods of this work, in order to facilitate reproduction in open source?
>
> We are adding a section to the appendix containing a simple, self-contained implementation of Edit Flows.
>
> > In Table 6, what were the CFG values tuned on? Train, train and validation, or validation only?
>
> The benchmarks do not have train-validation splits. We tested CFG values 0.0, 0.5, 1.0, 2.0, 5.0, and 10.0 on each benchmark and report the best results. No other hyperparameters were tuned.
>
> > On figure 7, do the authors have any comparison of parallelization versus sampling steps to fit a given computational budget, similar to "Large Language Monkeys: Scaling Inference Compute with Repeated Sampling"? For example, is it better to do pass @64, with smaller steps, or pass @10 with more steps?
>
> Thank you for bringing this work to our attention. The technique is certainly applicable, and it can be used to optimize the sample quality at a given compute cost. We did not investigate sampling efficiency in this submission, but we believe this is a worthwhile idea to improve inference cost and we hope to improve on this aspect in future work.
>
> > Similar to the above, reverse rates were discussed in the paper, but I did not see any concrete study of result quality over compute with reverse rate enabled, versus more parallelism or increased number of steps - do the authors have any results on this?
>
> First note that learning the reverse rate does not cost extra compute as it is learned simultaneously with the forward rate. The reverse rate is an extra output head with the same loss but time-inverted, incurring no additional compute cost during training.
>
> With the reverse rates disabled during sampling results in HumanEval+ pass@1, we observe a 2.2% drop for Edit Flow and a 1.9% drop for Uniform $X_0$ + Edit Flow.
>
> > One concern I have on this approach is vocabulary scaling beyond 32k.
>
> Scaling is analogous to that of standard Transformer models. Using a larger vocabulary size results in larger output and embedding layers, but does not affect the intermediate transformer blocks.
>
> > Do the authors have any qualitative examples of injecting invalid text (false facts, code vulnerability, typos, etc) during the sampling chain, and seeing if the model returns to edit and remove these? The space of human-in-the-sampling-chain seems possible given the model setup, depending on this behavior. Similarly, given it should be possible to mark a subsequence "not for editing" - does having some subsequence marked fixed change the following sampling behavior?
>
> We are adding an example to the appendix to showcase the model’s correction capability. Edit Flows are able to fix an incorrect implementation of the `is_prime` function (where the three return statement are negated). We start with the string
> ```
> def is_prime(n: int) -> bool:
>     if n <= 1:
>         return True
>     for i in range(2, n):
>         if i % n == 0:
>             return True
>     return False
> ```
> Over the course of 300 steps, the model makes 117 edits and reaches the final state
> ```
> def is_prime(n: int) -> bool:
>     if n <= 1:
>         return False
>     for i in range(2, n):
>         if n % i == 0:
>             return False
>     return True
> ```
> Note that the model does not make the minimal number of edits. For example, the state at step 150 contains a few extra random tokens which are deleted later in the sampling process:
> ```
> def is_prime(n: int) -> bool:
>     if n <= 1:
>         return False
>     for i in range(2, n):
>         if. % n == 0:
>             return False
>
>
>  True
>
>
> ,
> 6
> ```
> Note that these are sampled from the model while disabling edit operations on the function signature. Likewise, we can also disable editing in any subsequence at sampling time, but our model does not take such a mask as input and would not change its behavior apart from seeing different noisy sequences. As shown above, however, it is fine to prompt the model with invalid or wrong inputs and the model can still "denoise" the inputs.
>
> > Any intuition on CFG on rates versus CFG on Bregman divergences and when to prefer one or the other?
>
> In general, it is unclear how CFG affects the likelihood of the generated samples. This is the case for continuous-space diffusion models and is also the case for Edit flows. Only for the mask construction, it can be shown that CFG is equivalent to sampling from a similarly modified likelihood --- which corresponds to both CFG on rates and CFG on Bregman divergences (ELBO) as they are equivalent. Out of the two options for Edit Flows, the CFG on the Bregman divergence loss / ELBO is slightly more interpretable.
>
> List of manuscript changes in response to the feedback:
> 1. Results table names the specific autoregressive model (Llama3) and the superscript notation is now consistent.
> 2. Added a qualitative example to the appendix to showcase how Edit Flows can fix an incorrect implementation of the `is_prime` function.
> 3. Added qualitative examples to showcase the Mask DFM and Edit Flow generation processes.

---

> > ### Comment · Reviewer_arpi · 2025-08-04
> > **Response to the authors**
> >
> > Thank you for the detailed reply, this really answered all of my main questions and concerns, and I have updated my score to match. The inclusion of pseudo code in the appendix of the paper will go a long way to enabling follow-up work, and a number of my detailed questions have clear answers, as well as showing gains over existing benchmarks during the short-turnaround ablations I asked for.
> >
> > I would be curious in general about various sampling schemes (such as typical sampling) to deal with the "[object Object]" problem described here, but this is very strictly not necessary for this work to be accepted as it stands now. Thank you again to all authors, for their work on the paper and detailed reply to all reviewers.

---

### Decision · Program_Chairs · 2025-09-17

**Decision:**

Accept (poster)

**Comment:**

This paper introduces Edit Flows, a discrete flow matching framework for non-autoregressive sequence generation that uses edit operations (insertions, deletions, substitutions) rather than masking. The work addresses the limitation of fixed-length generation in existing non-autoregressive models by enabling variable-length sequence generation through a continuous-time Markov chain formulation. Strengths include strong theoretical foundations with rigorous mathematical framework, novel modeling approach that naturally handles variable-length sequences, comprehensive experimental evaluation across image captioning, text generation, and code generation tasks, and consistent improvements over masked diffusion baselines. The paper demonstrates 3x computational efficiency during training compared to masked approaches and includes interpretable generation processes with self-correction capabilities. However, weaknesses include performance still lagging behind autoregressive models on text/code tasks, high inference cost requiring 5K-10K sampling steps, limited scalability evaluation (models only trained up to 1B parameters), and some empirical results showing modest improvements with relatively low absolute performance on certain benchmarks.

The rebuttal period generated substantial discussion, particularly around empirical validation concerns. Reviewer arpi initially raised questions about baseline comparisons and efficiency claims but was satisfied by the authors' detailed responses, including additional experiments with Llama-3 initialization and clarifications about computational efficiency, ultimately increasing their score to "Strong Accept." Reviewer C6NG asked about computational fairness and theoretical details, and after receiving comprehensive explanations about the 3x efficiency gain and localized propagation paths, also moved to "Strong Accept." Reviewer LxY2 maintained their "Borderline Accept" after authors addressed concerns about implementation availability and sampling efficiency. However, reviewer czyg remained unconvinced despite extensive exchanges, raising persistent concerns about statistical significance of results, appropriateness of confidence interval calculations, and fundamental questions about the method's practical value. The reviewer argued that confidence intervals were incorrectly computed by treating multiple runs as independent trials rather than using bootstrap resampling, and questioned whether modest improvements on benchmarks with relatively low absolute scores constitute meaningful progress.

Despite czyg's remaining concerns about empirical validation which I resonate with, the strong theoretical contributions, novel approach to variable-length generation, and positive reception from three other reviewers support acceptance, as the work advances our understanding of non-autoregressive generation methods and provides a valuable foundation for future research.